# Hyperbranched polymer functionalized flexible perovskite solar cells with mechanical robustness and reduced lead leakage

Zhihao Li[1,2,3,4], Chunmei Jia[1], Zhi Wan[1], Jiayi Xue[1], Junchao Cao[2,3], Meng Zhang[1,2], Can Li[1,4], Jianghua Shen[2,3], Chao Zhang ⬡[2,3,5] ✉ & Zhen Li ⬡[1,4] ✉

Perovskite solar cells (PSCs) are multilayer structures. The interface between electron transport layer and perovskite is the mechanical weakest point in flexible PSCs due to its low fracture energy. Herein, we develop a highly adhesive polyamide-amine-based hyperbranched polymers to reinforce the interface. The interface fracture energy is improved from 1.08 to 2.13 J·m$^{-2}$ by the hyperbranched polymers with adhesive groups and dynamic hydrogen bond networks. The polymer functionalized perovskite solar cells achieve superior power conversion efficiencies of 25.05% and 23.86% for rigid and flexible devices, respectively. Furthermore, the hyperbranched polymer contains abundant intramolecular cavities that can capture Pb$^{2+}$. Pb leakage after solar cell damage is effectively suppressed. Our findings provide insights on designing adhesive interface layers towards high-efficiency, mechanical-stable and environment-friendly flexible perovskite solar cells.

Metal halide perovskite solar cells (PSCs) have emerged as a highly promising photovoltaic material, characterized by a high absorption coefficient[1], tunable band gaps[2], long carrier diffusion length[3,4], and low exciton binding energy[5,6]. The progress made in PSC technology is evidenced by the record power conversion efficiency (PCE), which has surged from 3.8% to a certified 26%, placing it in competition with Si solar cells[7–12]. One of the most attractive features of PSCs is their low cost and solution-based processing, which makes them more versatile than conventional photovoltaic technologies[13–16]. These advantages have positioned perovskites to compete with incumbent solar technology on a global scale[17,18]. Nevertheless, the mechanical properties of PSCs have been shown to play a critical role in device yield during manufacturing[19–21]. Despite a focus on the efficiency, large-area

PSCs[22,23], and scalable manufacturing[24–26], the failure modes related to mechanical stresses remain largely unexplored, limiting the reliability of PSCs. As such, the tipping point for perovskite commercialization depends on improving the mechanical reliability of PSCs.

The PSC is a multilayer structure, with each layer having distinct mechanical properties. Previous studies have revealed that the adhesive fracture energy ($G_c$) of PSCs ($G_c < 1.5$ J m$^{-2}$) is inferior to that of organic ($G_c$-5–15 m$^{-2}$), CIGS ($G_c$-10 m$^{-2}$) and c-Si solar cells (-10–200 m$^{-2}$)[27,28]. Specifically, the interface between electron transporting layers (ETLs) and perovskite layer is particularly fragile. The formation of small cracks during bending may propagate into larger ones during subsequent cycles, leading to delamination fractures during repeated bending and poor long-term bending

[1]State Key Laboratory of Solidification Processing, Center for Nano Energy Materials, School of Materials Science and Engineering, Northwestern Polytechnical University and Shaanxi Joint Laboratory of Graphene (NPU), Xi'an 710072, P. R. China. [2]Department of Aeronautical Structure Engineering, School of Aeronautics, Northwestern Polytechnical University, Xi'an, Shaanxi 710072, China. [3]Shaanxi Key Laboratory of Impact Dynamics and Its Engineering Application, Joint International Research Laboratory of Impact Dynamics and Its Engineering Applications, Xi'an, Shaanxi 710072, China. [4]Research & Development Institute of Northwestern Polytechnical University in Shenzhen, Shenzhen 519057, China. [5]School of Civil Aviation, Northwestern Polytechnical University, Xi'an 710072, China. ✉e-mail: chaozhang@nwpu.edu.cn; lizhen@nwpu.edu.cn

cyclability[29–31]. The weak interface binding energy represents a critical bottleneck that deteriorates the integrity and device performance of flexible PSCs.

Significant efforts have been directed towards mitigating the interfaces delamination issue in PSCs through interface modifications[32,33], such as deploying interfacial buffer layers[34–36] and constructing interpenetrating scaffold interfaces[37,38]. Ordered dipolar structure of β-poly(1,1-difluoroethylene) (β-pV2F) was used to create a self-assembly polymeric layer with strain-buffering and lattice-stabilizing effect to mitigate the tension and compression strain in perovskite during thermal cycling, achieving an excellent operational stability of PSCs[39]. Polystyrene (PS) was used as an interface buffer to reduce residual stress during annealing, thereby reducing interface defects and enhancing device performance[40]. An adhesive polymer poly(ethylene-co-vinyl acetate) (EVA) was introduced between $SnO_2$ and perovskite, improving the interface contact and increase fracture toughness, resulting in better performance and mechanical stability[41]. However, these interface modification methods typically employed small molecule or linear polymers, which offered only limited interface strength and ductility. Inspired by the natural materials[42–47], dynamic and reversible non-covalent bonds have gained significant attention for enhancing material toughness and introducing self-healing and anti-fracture properties. Compared to covalent bonds, the dynamic non-covalent can effectively dissipate energy via reversible bond breaking and formation or bond exchange reactions, resulting in highly stretchable materials.

Hyperbranched polymers (HBPs) are a distinct class of polymers featuring branched three-dimensional (3D) molecular structure, abundant functional end groups and nanometer-size intramolecular cavity. Tuning the molecular structure and functional end groups enables precise control over the physical and chemical properties of HBPs[48,49]. Introducing complementary hydrogen bond donor and acceptor into the branches of HBPs creates an internal dynamic bonding network, which can absorb mechanical energy and facilitate

healing of micro-fracture[50]. The globular shape and intramolecular cavities of HBPs contributes to a high mobility of the branches, enabling substantial molecule deformation. These unique configurations provide HBP materials with exceptional mechanical strength and toughness[51,52]. In addition, polar end groups can form robust interactions with other materials, imparting exceptional adhesive characteristics. As a consequence, elastic adhesive HBPs have attracted significant attention as promising interface enforcement materials for diverse optoelectronic devices[53,54].

Lead toxicity is a significant concern in the commercialization of PSCs. The issue of lead leakage would face more scrutiny in flexible PSCs, as these devices are intended for use in commercial electronics and wearable devices. Encapsulants have been developed to control the risk of lead leakage when PSCs are damaged or broken[55,56]. Typically, these materials contain large amounts of functional groups that bind with soluble $Pb^{2+}$. HBPs with $Pb^{2+}$ binding groups can be deliberately designed to act as an internal encapsulant, mitigating lead leakage and creating a safer condition for their daily use[57].

Herein, we synthesized series of polyamide-amine-based HBPs with varying branch lengths. The HBPs possess high density of amide, primary amine, secondary amine and carboxyl groups at the spherical surface and nanometer-sized intramolecular cavity (Fig. 1a). These highly polar groups present strong external hydrogen bonds with $SnO_2$ and perovskite, resulting in strong adhesion at the fragile ETL/perovskite interface. The high mobility of the HBPs branches facilitates regrouping of the hydrogen bond donors and acceptors, creating a self-adaptive dynamic hydrogen bond network. Simultaneously, the intramolecular cavities can undergo structural reconfiguration during bending and absorb deformation energy. These unique properties allow the HBPs to effectively absorb and alleviate deformation energy, thereby enhancing the mechanical stability of flexible PSCs. Our results show that rigid PSCs based on HDA-HBPs modified $SnO_2$ exhibit superior PCE up to 25.05%. While flexible PSCs exhibit PCE up to 23.86% with excellent mechanical flexibility, maintaining 88.9% of their

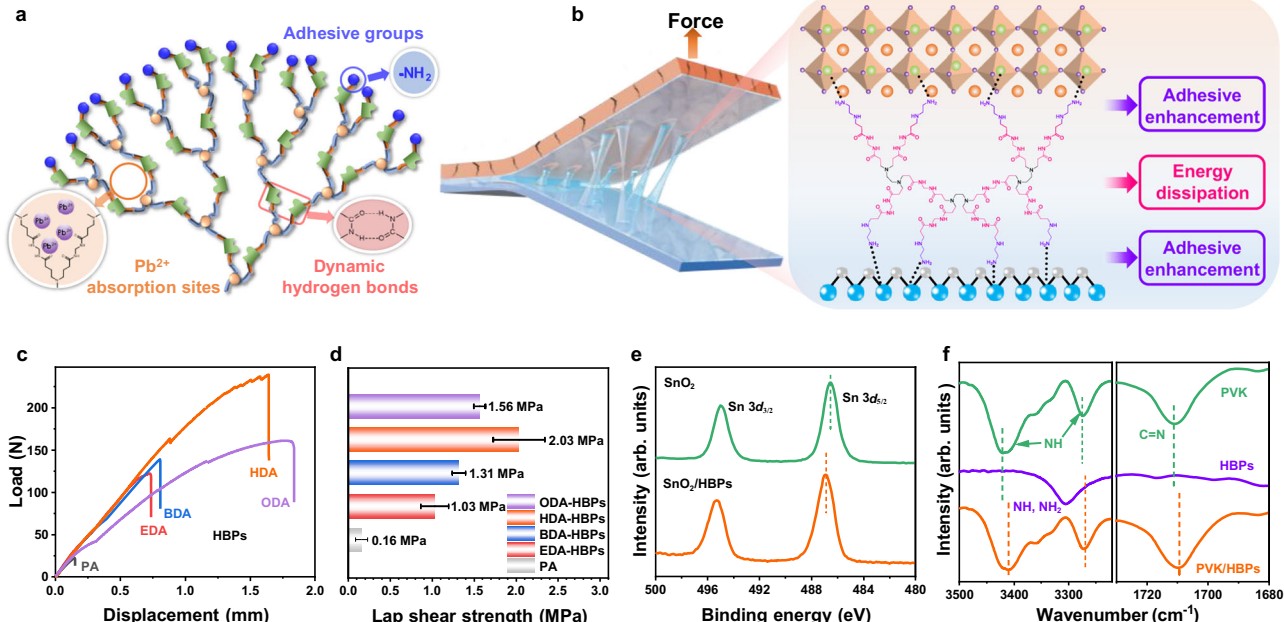

**Fig. 1 | Bonding effect of hyperbranched polymers (HBPs) adhesive layer to $SnO_2$ and perovskite layers. a** Molecular structure of HBPs. **b** Schematics of the HBPs bonding between $SnO_2$ and perovskite interface. **c** Lap shear curve and **d** adhesion strength of HBPs bonding to ITO substrates. Error bars represent the standard deviations from the adhesion strength results of five bonding materials. **e** X-ray photoelectron spectra (XPS) of Sn 3d peaks of $SnO_2$ and HBPs-modified $SnO_2$ films. **f** Fourier-transform infrared (FTIR) spectra of control and HBPs-modified perovskite films. *J. Mater. Chem. B.* 9, 585–593 (2020). Source data are provided as a Source data file.

initial PCE after 10,000 mechanical bending cycles at a bending radius of 3 mm. Moreover, the abundant carboxyl and amine groups present in the intramolecular cavity of HBPs have the capability to bind with $Pb^{2+}$, substantially mitigating the risk of lead leakage even after structure failure of the flexible PSCs.

## Results and discussion

### Bonding effect of HBPs adhesive layer to $SnO_2$ and perovskite layers

The synthesis of HBPs was carried out using a one-pot Michael addition reaction between N,N′-methylene diacrylamide (MBA) and different amines, as shown in Fig. 1a and Supplementary Fig. 1. The molecular structure was confirmed using [1]H nuclear magnetic resonance ([1]H-NMR, Supplementary Fig. 2) and Fourier transform infrared (FTIR, Supplementary Fig. 3) spectroscopy. Four different alkyl spacers with varying chain lengths were used to investigate the influence of the chain flexibility on mechanical properties of HBPs. These HBPs were named EDA-HBPs, BDA-HBPs, HDA-HBPs, and ODA-HBPs, respectively, for ethylenediamine, 1,4-butanediamine, 1,6-hexanediamine, and 1,8-octanediamine derived HBPs. The polar $-NH_2$ end groups in the HBPs can form high-density multiple hydrogen bonds with $SnO_2$ and perovskite layers, which are beneficial to provide greater adhesive strength[58,59]. The inter- and intramolecular H bond interactions between the -NH-, $-NH_2$, and C=O groups create a dynamic hydrogen bond network with strong mechanical strengths, preventing crack propagation and increasing the toughness of the adhesive interface (Fig. 1b)[60].

The mechanical property of the HBP is determined by its molecular structure. We studied the influence of HBPs molecular configuration on the H-bond formation ability. HBPs with longer branches have more loose structure and characteristics of higher elasticity and flexibility. This is because the longer aliphatic chains allow for sufficient single bonds to rotate internally, reducing steric hindrance during synthesis and leading to a larger molecular size with more loose structure (Supplementary Fig. 4). Differential scanning calorimetry (DSC) curves of the HBPs are shown in Supplementary Fig. 5. HBPs with different repeating units show different glass transition temperature ($T_g$) due to the segment moving ability affected by intramolecular interactions. The HDA-HBPs possessed the highest $T_g$ due to the fact that the HDA-HBPs have more branches with functional groups that lead to more hydrogen bonds, which brings the polymer backbones closer, thus making the material stronger and more heat resistant (Supplementary Fig. 6). The intramolecular hydrogen bonding between the amino and carbonyl groups in HBPs was confirmed by the shift of FTIR absorbance peak to a lower frequency, as shown in Supplementary Fig. 7. The dynamic hydrogen bond is also demonstrated by FTIR band shift of the amide, −NH− and $−NH_2$ groups in associated state to free state at raising temperature from room temperature to 175 °C (Supplementary Fig. 8). Only HDA-HBPs and ODA-HBPs showed crystalline peaks in XRD, and HDA-HBPs exhibited sharper peaks (Supplementary Fig. 9). The crystalline micro-domains in HDA-HBPs can improve the mechanical strength of the material and benefit to the adhesive strength.

Lap-shear tests were conducted to quantitatively evaluate the adhesion strengths of HPBs, as depicted in Fig. 1c. The results revealed that HDA-HBPs exhibited the highest adhesive strength of $2.03 \pm 0.3$ MPa (Fig. 1d). The 3D spatial configuration of HBPs enhanced the interaction between neighboring molecule chains, enhancing the crosslinking strength and mechanical robustness of the elastomeric adhesive. Furthermore, globular and dendritic molecular structure of HBPs conferred low crystallinity and less chain entanglement, resulting low viscosity and high solubility, in compare to linear polyamide (PA) with similar molecular weights and composition. Linear PA molecule tend to entangle and aggregate owing to the high polarity of the amide group. As a consequence, PA exhibited larger particle sizes in solution

compared to HBPs (Supplementary Fig. 10). The entanglement of polymer chain and segment aggregation restricted the mobility of the PA polymer chains, leading to an inferior mechanical strength and adhesive behavior. Given its well-balanced blend composition and superior adhesion properties, HDA-HBPs were selected as the optimal HBPs for constructing the interfacial layer. Henceforth, we will refer to HDA-HBPs as HBPs for simplicity in the subsequent discussions.

To gain further insight into the chemical interactions between HBPs and the adjacent layers, we conducted X-ray photoelectron spectroscopy (XPS) characterizations. As shown in Fig. 1e, the binding energies of the Sn $3d_{5/2}$ (486.5 eV) and $3d_{3/2}$ (495.0 eV) peaks of $SnO_2$ were upshifted to 486.9 eV and 495.3 eV after HBPs modification, indicating a strong chemical interaction between HBPs and $SnO_2$. The presence of additional C-N (Supplementary Fig. 11b) and N $1s$ (Supplementary Fig. 11c) peaks in XPS further confirmed the existence of HBPs at the surface of the $SnO_2$ ETL. In addition, the O $1s$ spectra in Supplementary Fig. 11d showed that the ratio of saturated oxygen ($O^{2-}$) to hydroxyl group (-OH) in $SnO_2$ increased after HBPs modification, suggesting reduction of surface defect density (Supplementary Table 1). FTIR spectroscopy was also used to analyze the HPBs interactions with $SnO_2$ ETLs (Supplementary Fig. 12). The characteristic peak of HBPs at 1638 $cm^{-1}$ (-C=O) shifted to 1645 $cm^{-1}$, and the -NH-, $-NH_2$ peaks at 3305 $cm^{-1}$ shifted to 3299 $cm^{-1}$. The peak of Sn-O-Sn shifted from 681 to 698 $cm^{-1}$, indicating that the -C=O, -NH-, and $-NH_2$ groups of HBPs can passivate the unsaturated Sn dangling bonds, which is consistent with the XPS results[61].

The chemical interaction between HBPs and PVK was also characterized using XPS. The Pb $4f_{5/2}$ and Pb $4f_{7/2}$ peaks (Supplementary Fig. 13a) shifted to higher binding energy, while the O $1s$ (Supplementary Fig. 13b) peak moved toward lower binding energy, indicating strong chemical interaction between Pb and O with the presence of electron-rich C=O groups in the HBPs. Moreover, the two additional peaks of metallic Pb(0) at 141.2 and 136.2 eV disappeared after HBPs modification, suggesting effective passivation of the uncoordinated Pb atoms. In addition, the FTIR spectrum (Fig. 1f) revealed that the stretching vibration band of C = N and -NH in perovskite formed hydrogen bonds with the -NH- and $-NH_2$ groups of HBPs, resulting in a slight shift of about 15 $cm^{-1}$ toward lower wavenumber, which is consistent with the shift of N $1s$ and I $3d$ peaks in the XPS spectra (Supplementary Fig. 13c, d). Therefore, it can be inferred that HBPs can form chemical interactions with both $SnO_2$ and perovskite layers through hydrogen bonds and coordination bonds, thus acting as a bridge between ETL and perovskite and strengthen the interface contact.

### PSC performance with HBPs modification

To investigate the impact of HBPs modification on photovoltaic performance of the PSCs, we fabricated PSCs with the architecture of ITO/ $SnO_2$/(HBPs)/perovskite/spiro-OMeTAD/Au, as illustrated in Fig. 2a. The HBPs modification resulted in an improved solar cell performance. The optimal HBPs concentration was attained from the current density–voltage ($J-V$) curves in Supplementary Fig. 14. The impact of HBPs chain length on device performance was also investigated and presented in Supplementary Fig. 15. Among all the HBPs, HDA-HBPs exhibited the best device performance, possibly due to the stronger interaction between HBPs and the adjacent $SnO_2$ and perovskite layers. The control device showed a PCE of 21.14%, with $V_{OC} = 1.119$ V, $J_{SC} = 24.16$ mA $cm^{-2}$, FF = 0.78. The champion PSCs with HDA-HBPs demonstrated a PCE of 25.05% and negligible hysteresis, with $V_{OC} = 1.167$ V, $J_{SC} = 25.60$ mA $cm^{-2}$, and FF = 0.84 at reverse scan (Fig. 2b). The $V_{OC}$, $J_{SC}$, FF, and PCE statistics were provided in the supporting materials (Supplementary Fig. 16 and Supplementary Table 2), showing a higher and narrower distribution of PCE in HBPs modified devices. The average PCE of 20.50% for control devices increased to 24.39% for HBPs-modified devices (Fig. 2c),

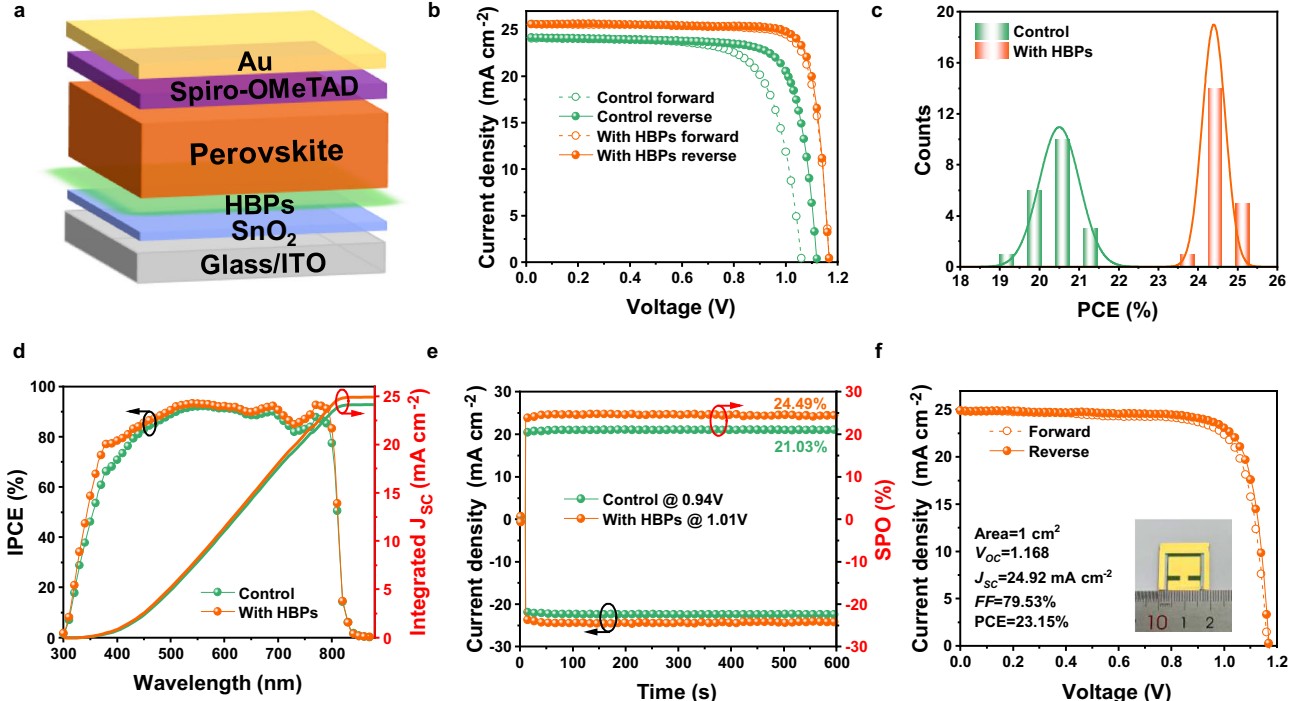

**Fig. 2 | Perovskite solar cell performance with HBPs modification. a** Device architecture of the HBPs-modified PSCs. **b** *J–V* curves of the champion devices with and without HBPs. **c** The PCE histogram of PSCs. **d** External quantum efficiency (EQE) spectra and integrated current of the PSCs. **e** Stabilized photocurrent and power output of PSCs. **f** *J–V* curve of the champion HBPs-modified PSCs with large area of 1 cm². Inset shows a photo of the large area solar cell. Source data are provided as a Source date file.

demonstrating PCE improvement in a reproducible way. The integrated current density obtained from the external quantum efficiency (EQE, Fig. 2d) measurements were 24.15 and 24.93 mA cm⁻² for the control and HBPs-modified devices, respectively, which match well with the $J_{SC}$ attained from the *J-V* measurements. The stabilized photocurrents and power outputs were measured at the maximum-power-point (MPP) voltage and shown in Fig. 2e. Stabilized PCEs of 24.49% and 21.03% were obtained for the control and HPB-modified PSCs, respectively. Moreover, large-area (1 cm²) devices were fabricated and a champion PCE of 23.15% was achieved (Fig. 2f), demonstrating the great potential of deploying HBPs in large-area PSC fabrication. Linear-shaped polymer adhesive polyamide (PA) was also chosen as a model modification polymer to compare with the HBPs. The optimal concentrations and the best PCE with PA were attained from Supplementary Fig. 17. The PA-modified PSCs showed lower PCE of 22.4%, although PA has the same amount of amide groups as HBPs. HBPs endow spreading branches with less entanglement, exposing more passivation groups and enhancing the interaction with the SnO₂ and perovskite layers. Therefore, HBPs offer more improvement to the photovoltaic performance of PSCs.

## Characterizations of the functionalized SnO₂–perovskite interface

UV-vis transmittance spectroscopy (Supplementary Fig. 18) and atomic force microscopy (Supplementary Fig. 19) show that the HBPs interface layer has little influence on the light transmission and surface roughness of the SnO₂ ETLs. Furthermore, due to the increased adhesion between the perovskite and SnO₂ ETL with HBPs modification, the quality of perovskite film improved, including better surface coverage, larger grain size, and higher crystallinity. As shown in Supplementary Fig. 20, the perovskite film based on HBPs-modified SnO₂ exhibits compact morphology, with the average grain size increasing from 543 nm to 895 nm, which is consistent with a slightly increased

UV-vis absorption (Supplementary Fig. 21) and thus contributes to a slight increase in $J_{SC}$. Additionally, better interface quality can be observed from the cross-sectional SEM images in Supplementary Fig. 22. The perovskite films prepared on HBPs-modified SnO₂ exhibit large vertically oriented grains that spans the entire absorber layer. There is no cavity found between the perovskite layer and HBPs-modified SnO₂ ETL, which is beneficial for electron carrier extraction and environmental stability. Furthermore, the linear Williamson-Hall plot of the XRD data from Supplementary Fig. 23 characterizes the variation in lattice strain of perovskite. The micro-strain of HBPs-modified perovskite film decreases to $1.21 \times 10^{-6}$ compared to the control film of $6.92 \times 10^{-5}$. Furthermore, the residual tensile strain of perovskite films was measured by grazing incident X-ray diffraction (GIXRD). The residual tensile strain of HBPs-modified was 28.30 MPa, constituting ~56% strain reduction compared to 64.24 MPa of the control sample (Supplementary Fig. 24). Due to the low glass transition temperature (39.8 °C), the HBPs can remain in a soft state to accommodate the thermal expansion mismatch between the perovskite layer and substrate cooling from the annealing temperature (150 °C), thus reducing residual strain during perovskite grain growth[40].

Steady-state photoluminescence (PL) and time-resolved photoluminescence (TRPL) measurements were carried out to understand the effects of HBP modification on carrier dynamics and charge recombination at the perovskite/ETL interfaces, as shown in Fig. 3a. The steady PL intensity of the perovskite decreased after HBPs modification, consistent with the reduced TRPL lifetime from 89.4 ns for the control to 12.7 ns for the HBPs-modified sample (Fig. 3b and Supplementary Table 3), indicating enhanced electron extraction and transport at the SnO₂/perovskite interface. Additionally, the perovskite deposited on the HBPs-modified glass substrates showed higher PL intensity and a slight blue-shift of the excitation peak, as confirmed by the increased PL lifetime from 119.6 to 489.9 ns (Supplementary Fig. 25 and Supplementary Table 4). We further investigate the electrical

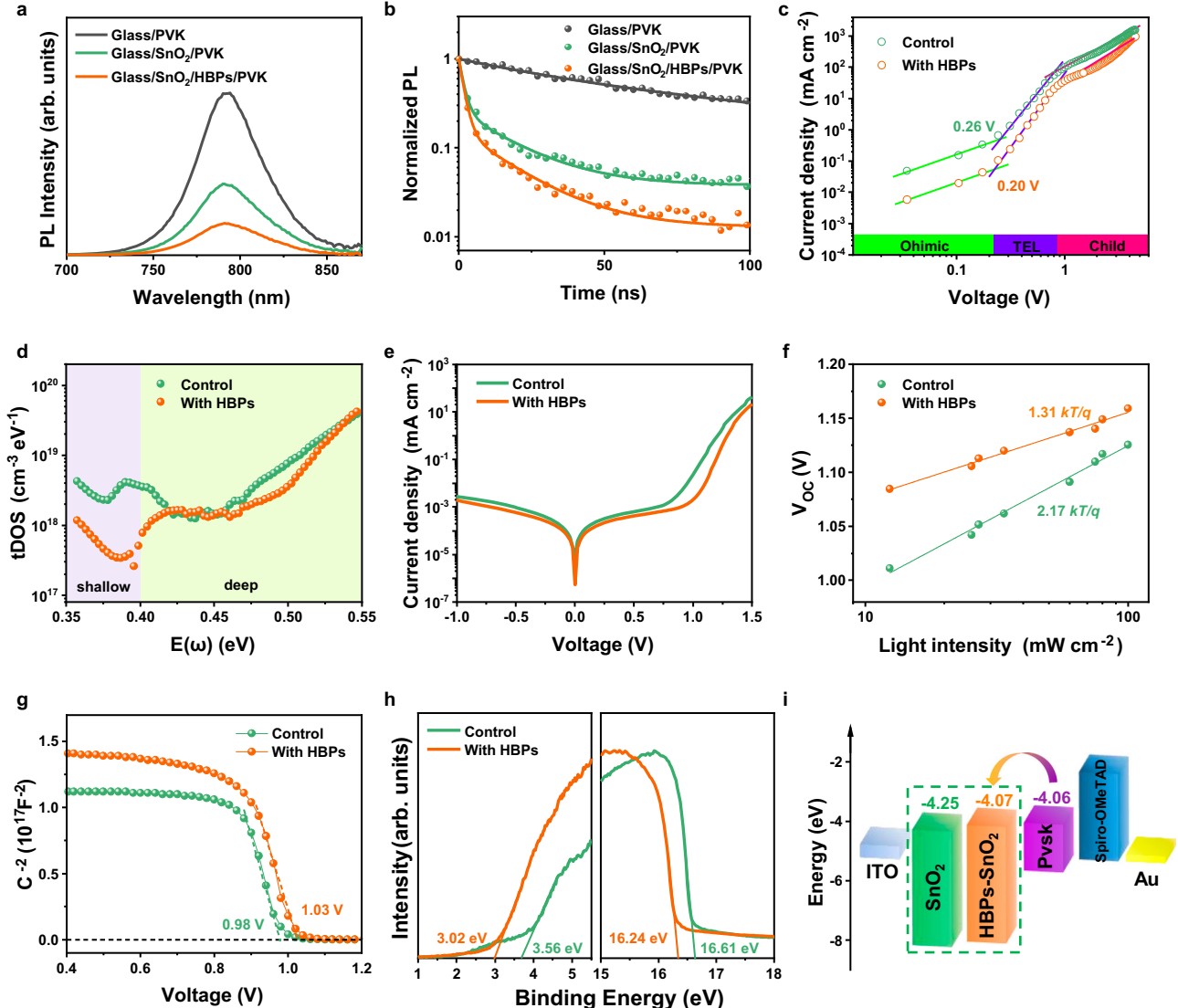

**Fig. 3 | Characterization of perovskite films and solar cells based on pristine SnO₂ and HBPs-modified SnO₂. a** Steady-state PL and **b** TRPL spectra of perovskite films on pristine SnO₂ and HBPs-modified SnO₂. **c** SCLC measurements of perovskite films with the electron-only structures. **d** tDOS of devices based on SnO₂ with and without HBPs modification. **e** Dark $J$–$V$ curves of PSCs. **f** $V_{OC}$ as a function of light intensities of PSCs. **g** Mott−Schottky plots and **h, i** UPS and energy diagram of PSCs with and without HBPs modification. Source data are provided as a Source Date file.

properties of the buried interfaces (Supplementary Fig. 26). Higher surface potential and elevated conductivity of the buried interface rigorously shows that the HBPs layer can improve the carrier extraction at the ETL/perovskite interface. This suggests that HBPs layer can effectively passivate surface trap states and suppress nonradiative recombination in the perovskite layer, ultimately resulting in improved $V_{OC}$ in PSCs. We further characterized the trap density of the perovskite films by space-charge limited current (SCLC) with an electron-only device structure (ITO/SnO₂/perovskite/PCBM/Au), as shown in Fig. 3c. The $V_{TFL}$ decreased from 0.26 V to 0.20 V after the HBPs modification, and the calculated trap density is $1.62 \times 10^{15}$ cm⁻³ for the HBPs-modified device, which is lower than the $2.11 \times 10^{15}$ cm⁻³ of the control device. The reduced trap density indicated that HBPs interlayer can effectively passivate defects and suppress charge carrier recombination, consistent with the PL and TRPL results. The trap density of states (tDOS) was further analyzed by thermal admittance spectroscopy (Fig. 3d). The shallower traps (0.35−0.4 eV) are corresponded to grain boundary traps and the deeper traps (0.4−0.52 eV) are originated from surface defects[62]. The perovskite film displayed a lower tDOS

value after HBPs modification, indicating that the HBPs can effectively passivate defects.

Dark $J$-$V$ measurements indicated that the HBPs modification improved the hole-blocking ability and suppressed charge recombination at the ETL/perovskite interface, as shown in Fig. 3e. In addition, the light-intensity-dependent $V_{OC}$ measurement (Fig. 3f) shows that the HBPs-modified device exhibits reduced trap-assisted recombination, as indicated by the smaller slope of 1.31 kT/q compared to 2.17 kT/q for the control device. The Mott-Schottky analysis (Fig. 3g) indicates that the HBPs-modified device has a higher built-in potential ($V_{bi}$) of 1.03 V compared to 0.98 V for the control device, which indicates a stronger built-in electric field between the SnO₂ and perovskite layer. The stronger electric field can enhance the charge separation and suppress carrier recombination, resulting in a higher $V_{OC}$.

The Fermi energy ($E_F$), valance band maximum (VBM), and conduction band minimum (CBM) extracted from the ultraviolet photoelectron spectroscopy (Fig. 3h) and Tauc plot (Supplementary Fig. 27) were summarized in Supplementary Table 5. The corresponding band

diagrams of the SnO₂/perovskite are illustrated in Fig. 3i. The CBM of the HBP-modified SnO₂ is −4.07 eV as compared to −4.25 eV of the pristine SnO₂. The higher CBM of the HBPs-modified SnO₂ is better aligned with the CBM of the perovskite, which reduces the interface energy barrier and facilitates electron transport. The higher CBM also creates a larger built-in potential, which pushes up the $V_{OC}$ of PSCs. Overall, these results indicate that the HBPs modification can effectively improve the charge carrier dynamics and reduce charge recombination at the perovskite/ETL interfaces, leading to higher $V_{OC}$ and improved device performance.

The adhesion strength between the SnO₂ and perovskite layers was tested using a diamond nano-indenter with a linearly increased load, as illustrated in Fig. 4a. The perovskite films after scratching test were shown in the SEM images. In the control film, with a continuous load, the probe penetrated the perovskite film, resulting in exposed ITO substrate (marked by blue dash line). The perovskite film with HPBs interface modification merely showed underneath ITO substrate, suggesting a stronger adhesion between the SnO₂ and perovskite layers. The interlayer failure points were detected by the critical load at which a jump in the contact acoustic emission signal was observed (Fig. 4b). The scratch force and adhesion strengths of the perovskite films with other polymers modified SnO₂ were presented in Supplementary Fig 28. All the HPBs showed stronger adhesion than linear polymer PA due to the branching and internal hydrogen bonds.

Lap-shear tests were utilized to quantify the adhesion strength between the SnO₂ and the perovskite layers (Supplementary Fig. 29).

The adhesives were applied onto the SnO₂ substrates and then hot-pressed onto the perovskite layer on a separate substrate. The substrates were then pulled apart to determine the adhesion strength at the point of failure. A higher lap shear strength corresponds to a stronger interface bonding. The control device exhibited clear delamination between the perovskite and SnO₂ layers, while the HBPs-modified device showed improved interfacial adhesion between the two layers. HDA-HBPs demonstrated the highest adhesion strength (0.86 ± 0.06 MPa), which confirms the strong adhesion ability of HDA-HBPs (Supplementary Fig. 29d). The fracture energy (G_c) of the SnO₂/(HDA-HBPs)/PVK interface was also measured using double cantilever beam (DCB) measurements, as illustrated in Fig. 4c and Supplementary Fig. 30. The HDA-HBPs modified SnO₂ interface exhibited a larger fracture energy of 2.13 J m⁻² compared to 1.08 J m⁻² of the pristine SnO₂ interface (Fig. 4d). Figure 4e provides a schematic representation of how is the HDA-HBPs affect the delamination process[63]. The unique structure of the HDA-HBPs interface layer can provide energy dissipation via reversible hydrogen bond networks to improve mechanical performance. With increasing tensile stress, polymer chains likely slipped and exchanged their H-bonded pairs to dissipate the fracture energy through dynamic dissociation and formation of H bonds, improving the interface toughness, thus resulting in the high G_c. SEM images of the delaminated surface of SnO₂ ETL and perovskite after DCB tests are shown in Fig. 4f, g, respectively. Some small perovskite grains were observed on the HBPs-modified SnO₂ surface, while the pristine SnO₂ had a relatively smooth fracture surface. The perovskite surface with HDA-HBPs

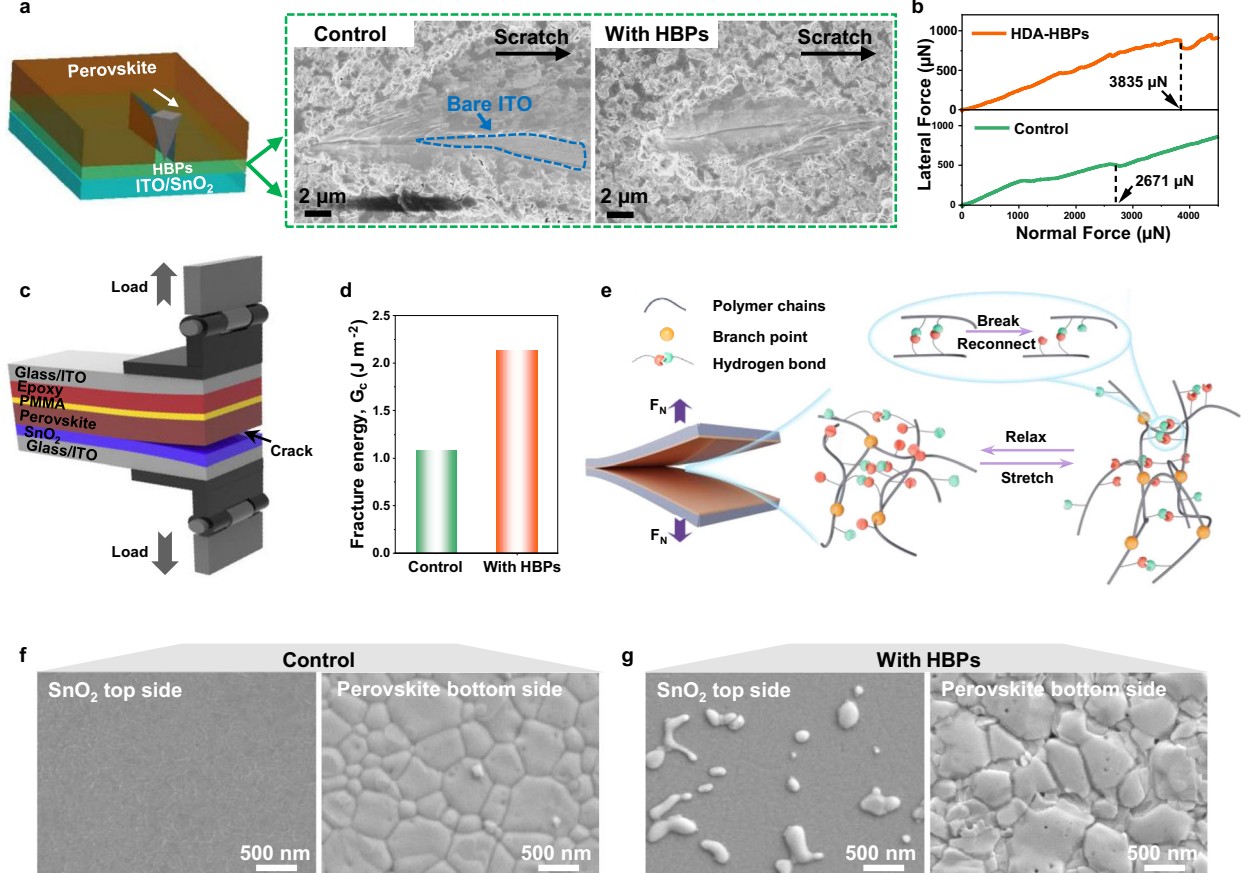

**Fig. 4 | Mechanical properties of the HPBs-modified interfaces. a** Schematic illustration of the nano scratch for adhesive strength testing and corresponding SEM images of perovskite films after nano scratch test. **b** Scratch curve of perovskite film. **c** Schematic illustration of double cantilever beam (DCB) test. **d** Fracture energy of interface with and without HBPs modification. **e** Schematic illustration for the energy dissipation mechanism at the ETL/PVK interface. **f, g** SEM images of fracture surface (SnO₂ ETL top side and perovskite bottom side) of the PSC with and without HBPs modification. *Science* 372, 618–622 (2021) *Adv. Mater.* 31, 1904029 (2019). Source data are provided as a Source Data file.

modification exhibited some holes left by the perovskite grains that attached to the SnO$_2$ surface. This again suggests an enhanced adhesion between SnO$_2$ ETL and perovskite layer via hydrogen and coordination bonds of HDA-HBPs. Interface reinforcement by HDA-HBPs can effectively prevent layer delamination, which is beneficial for the mechanical stability of the flexible PSCs.

## Flexible PSCs with HBP modification

Flexible PSCs were fabricated on polyethylene terephthalate/ITO (PET/ITO) substrates with HBPs adhesive layers and achieved a champion PCE of 23.86%, with $V_{OC}$ of 1.177 V, $J_{SC}$ of 25.07 mA cm$^{-2}$, and FF of 80.83%, as shown in Fig. 5a. In contrast, the control device achieved a best PCE of 20.60% ($V_{OC}$ = 1.086 V, $J_{SC}$ = 24.11 mA cm$^{-2}$, and FF = 78.66%). We have verified the PCEs using steady-state power output (SPO) measurements, and the stabilized PCEs are 23.16% and 20.27% for the F-PSCs with and without HBP layer, respectively (Supplementary Fig. 31). After modified with HBPs, the integrated $J_{SC}$ obtained from

EQE measurement increases from 23.55 mA cm$^{-2}$ to 24.06 mA cm$^{-2}$, which are consistent with the $J_{SC}$ obtained from the $J$-$V$ curves (Supplementary Fig. 32). The statistical photovoltaic parameters (PCE, $J_{SC}$, $V_{OC}$, and FF) of flexible PSCs were summarized in Supplementary Fig. 33 and Supplementary Table 6. Furthermore, we have also fabricated a large-area flexible device (1 cm$^2$) and obtained a champion PCE of 21.86% (Fig. 5b). The relatively lower PCE at large area is mainly attributed to the poor conductivity of ITO on flexible substrates, which deteriorates the FF of the device.

Lattice tensile strain can induce defects and nonradiative recombination centers in perovskite materials. In this study, we investigated the effects of HBPs modification on lattice strains by applying mechanical stress and utilizing the Williamson-Hall (W-H) plot derived from the XRD patterns (Supplementary Fig. 34, 35). As illustrated in Fig. 5c, the HBPs-modified device exhibited smaller changes of micro-strain during bending from $9.36 \times 10^{-4}$ to $9.42 \times 10^{-4}$ ($\Delta\varepsilon = 0.64\%$) compared to the control device (from $1.3 \times 10^{-3}$ to $1.54 \times 10^{-3}$,

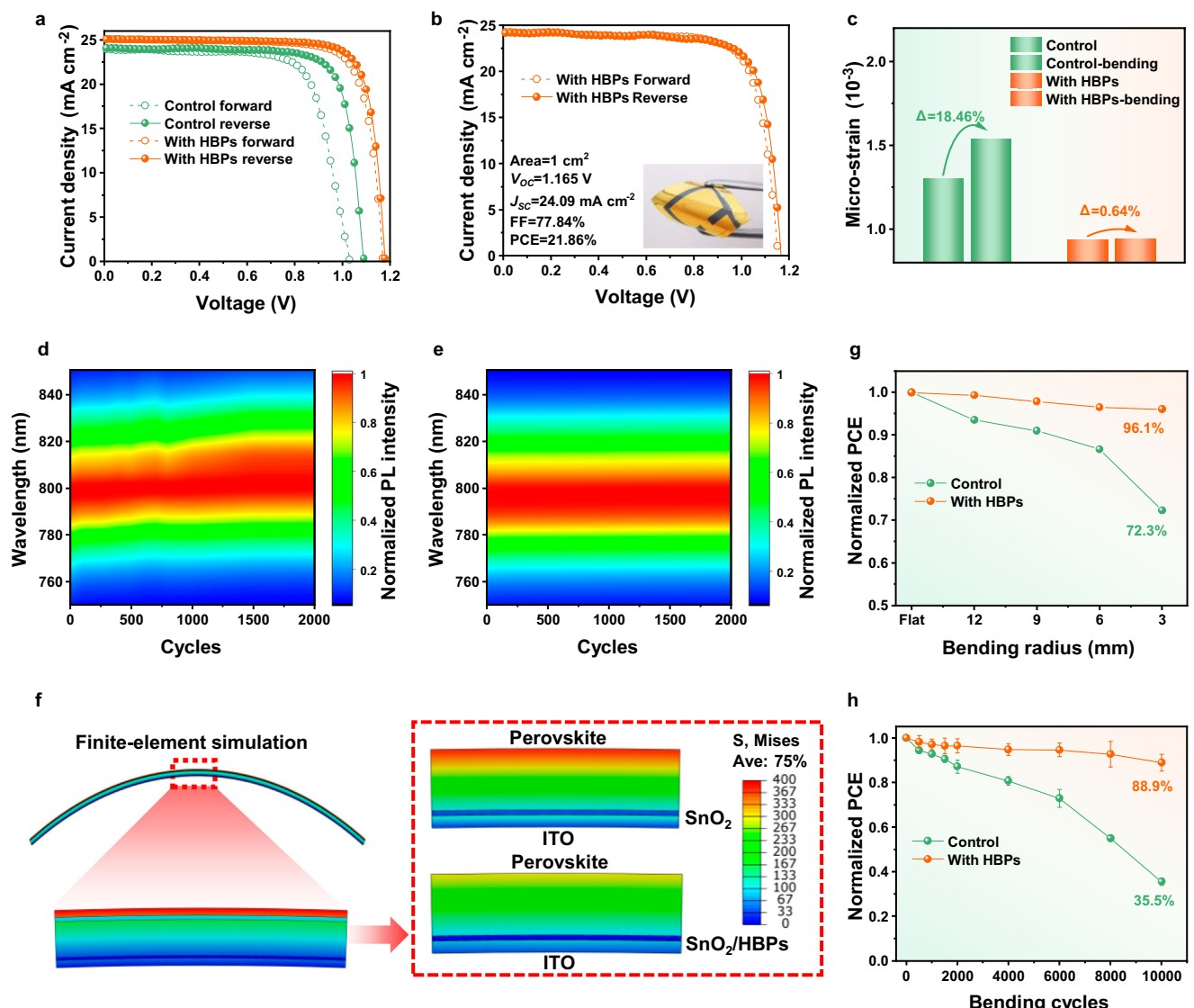

**Fig. 5 | Mechanical stability of flexible PSCs with HBPs modification. a** $J$-$V$ curves of the champion flexible devices with and without HBPs. **b** $J$-$V$ curve of the champion HBPs-modified flexible PSCs with an area of 1 cm$^2$. Insert is a photo of large-area F-PSCs. **c** Comparison of micro-strain of control and HBPs-modified perovskite films before and after bending. **d**, **e** Comparison of in-situ PL spectra of control and HBPs-modified perovskite films with 2000 bending cycles. **f** Finite-

element simulation of the devices. **g** Normalized PCE for flexible PSCs after bending at different curvature radii for 1000 cycles. **h** Normalized PCE for F-PSCs as a function of bending cycles with a bending radius of 3 mm. Error bars represent the standard deviations from the statistic results of two devices. Source data are provided as a Source data file.

$\Delta\varepsilon = 18.46\%$). The reduced micro-strain change indicates that HBPs can effectively relief local strain during tensile loading. To further evaluate the stability of the perovskite films after bending cycling, we conducted in-situ steady-state PL measurements on the perovskite films (Fig. 5d, e). It was found that the emission peak shift was almost undetectable for the perovskite film with HBPs-SnO$_2$ layer after 2000 bending cycles. In-situ PL were also conducted on perovskite films during tensile test (Supplementary Fig. 36). The average PL peak of the control film red-shifted from 787 to 800 nm when tensile strain increased from 0% to 10%, while the HPB-modified film only showed a slight redshift from 796 to 797 nm. Severe red shift of the peak position as well as peak broadening were observed in the control film after 100 stretching with strain of 2% (Supplementary Fig. 37). In contrast, the HBPs modified film exhibited almost no changes in the PL spectra after the repeatedly stretching test. Meanwhile, the trap density in control sample (42%) had a huge increase compared to the HBPs-modified sample (11%) (Supplementary Fig. 38). The improved stability of the film demonstrated the mechanical robustness of the film through HBPs modification, which can be attributed to the stress release at the SnO$_2$/perovskite interface by the soft HBPs interface layer. Furthermore, due to the low T$_g$ of HBPs, the hydrogen bond network is active at room temperature, allowing the material to self-adapt and healing to strain during bending and stretching, thus improving mechanical robustness and maintaining a stable perovskite composition and structure.

To further confirm the better mechanical stability of flexible PSCs via the adhesive HBPs interface layer, we measured the Young's modulus of the SnO$_2$ and perovskite layers using AFM mechanical model and nanoindentation techniques. The results, as presented in Supplementary Fig. 39, 40 and Supplementary Table 7, revealed that the Young's modulus of the SnO$_2$ and perovskite films decrease upon the introduction of HBPs interfacial layer. This decrease in Young's modulus leads to a reduction in the rigidity of each layer, which ultimately enhances the flexible device's resistance to bending. Furthermore, based on the measured mechanical properties, we performed finite element simulation to analyze the stress distribution of each layer in the flexible PSCs under bending. As illustrated in Fig. 5f, the HBPs adhesive layer significantly reduces the stress of the SnO$_2$ and perovskite layers, which could improve the mechanical durability of the device.

In addition, we evaluated the mechanical stability of F-PSCs at various curvature radii ($r = \infty, 12, 9, 6,$ and 3 mm), as depicted in Fig. 5g. The results demonstrate that as the bending radius decreased to 3 mm, the control device experienced significant degradation, with the PCE dropping to 72% of its initial value, while the flexible PSCs with HBPs maintained 96% of their initial PCE. Moreover, we carried out fatigue tests with a curvature radius of 3 mm and 10,000 bending cycles (Fig. 5h). The process and instrument of the fatigue testing are shown in Supplementary Movie 1. The flexible PSCs with HBPs interface layer exhibited excellent mechanical robustness and retain 88.9% of their original PCE after 10,000 bending, while the control device retained only 35.5% of its initial PCE. Furthermore, not like the perovskite film on pristine SnO$_2$, we did not observe any significant cracks on the perovskite film on the HBPs-modified SnO$_2$ after bending, thus substantiating the enhanced mechanical durability (Supplementary Fig. 41). Encapsulation-free flexible PSCs modified with HBPs maintained 89% of their original PCE after aging in ambient air with 30–40% relative humidity, whereas the control devices only retained 56% (Supplementary Fig. 42). Furthermore, under continuous 1-sun illumination in a nitrogen atmosphere, the HBPs-modified device retained 90% of its original PCE, whereas the efficiency of control device degraded to 67% (Supplementary Fig. 43). For the long-term operational stability, the efficiency of the control PSC decays to ~30% of the initial value, while the HBPs-modified PSC maintains 91% of its initial efficiency after continuous operation for 500 h (Supplementary Fig. 44). The significantly improved stability indicated that strain

relaxation in the perovskite through HBPs modification not only enhances the mechanical robustness but also improves long-term operational stability of the device.

## Lead absorption and sequestration by HBPs

To investigate the impact of using HBPs as internal encapsulation for mitigating lead leakage, we investigated the Pb$^{2+}$ absorption properties of HBPs. As shown in Supplementary Fig. 45 and Supplementary Table 8, the HBPs film showed a saturated adsorption capacity of 78.49 mg g$^{-1}$, higher than 53.53 mg g$^{-1}$ of linear polymer PA. The adsorption rate of HBPs film was 8.38 mg min$^{-1}$ g$^{-1}$, also higher than 4.75 mg min$^{-1}$ g$^{-1}$ of PA. The better Pb$^{2+}$ absorption properties can be attributed to the abundant internal cavity of HBPs. Subsequently, we encapsulated the bottom side and edges of the flexible PSCs with 10-μm-thick PET tapes (Supplementary Fig. 46). A 3 × 4 array of holes was intentional introduced by piercing through the flexible PSCs using a needle with diameter of 0.6 mm. The damaged PSCs were then immersed in 200 mL of deionized water to simulate lead leakage under immersion conditions. As depicted in Fig. 6a, yellow color appeared in the control device surrounding the needle-pierced holes and expanded in circular shapes within a few minutes. After 60 minutes, the yellow region expanded to most of the device area. In contrast, the device modified with HBPs only exhibited yellowing around the pierced hole area, and the yellow region did not spread outward. The lead concentration was determined using inductively coupled plasma mass spectrometry (ICP-MS). Within 30 minutes, the lead concentration in the water contaminated by the control device increased significantly due to lateral water penetration and the rapid dissolution of lead compounds. Remarkably, the device with HBPs modification retained a lead sequestration efficiency of 98% compared to the control device (Fig. 6b). This substantial reduction in lead leakage can be attributed to the following factors: (1) The less entangled polymer chains and intramolecular cavities of HBPs result in more exposed -C=O, -NH-, and -NH$_2$ groups, which can act as adsorption sites for Pb$^{2+}$. (2) The presence of strong hydrogen bond groups and hydrophobic polymer chain traps water molecules in the HBPs and effectively prevents lateral water penetration (Fig. 6c), as evidenced by the water diffusion test shown in Supplementary Figs. 47 and 48. The perovskite film was partly immersed into water. The water edge kept moving upward within 300 s in the control film, as compared to little change in the HBPs modified perovskite film. A water droplet was dropped onto the perovskite films and observed through optical microscope. The hydroscopic surface of the pristine perovskite cannot hold the water from spreading, and the area of the droplet kept expanding for 300 s. On the contrast, the HBPs construct a shielding to perovskite film, effectively preventing lateral water diffusion.

In summary, we have synthesized a range of polyamide-amine based HBPs with varying chain lengths and employed them as adhesive layers for flexible PSCs. The HBPs contain abundant polar groups, which strongly bind with the SnO$_2$ and perovskite layers, resulting in a strong interface adhesion. These polar groups also serve to passivate defects in both the SnO$_2$ ETLs and the perovskite layer, reducing interface recombination and facilitating carrier extraction. As a result, the HBPs-modified device exhibited champion PCE of 25.05% and 23.86% on rigid and flexible substrates, respectively. The HBPs possess complementary hydrogen bond donors and acceptors capable of slipping and exchanging of their H-bonds. The dynamic hydrogen bond network, together with the spherical molecular shape and nanometer-sized intramolecular cavities of HBP effectively dissipate fracture energy during deformation. Consequently, the HBPs reinforce the interface and increase the interface fracture toughness from 1.08 to 2.13 J m$^{-2}$. This leads to robust flexible PSCs that maintain 88.9% of their initial PCE after 10,000 bending cycles with a bending radius of 3 mm. Furthermore, the HBPs, with Pb$^{2+}$ absorption groups exposed on the molecular surface or within the intermolecular cavities, exhibit

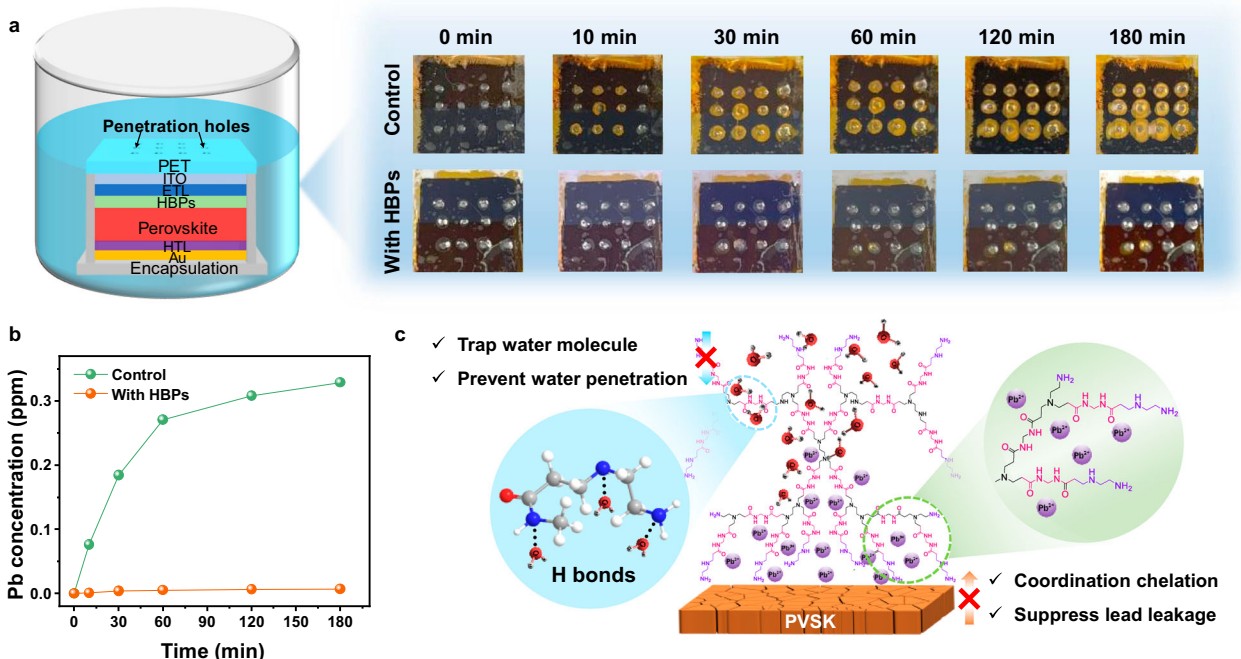

**Fig. 6 | Lead sequestration by HBPs. a** Schematics and photos of damaged flexible PSCs immersed in water for lead leakage test; **b** Pb concentration in the contaminated water measured by ICP-MS; **c** Illustration of the $Pb^{2+}$ sequestration mechanism. Source data are provided as a Source Data file.

enhanced $Pb^{2+}$ trapping ability, achieving a lead sequestration efficiency of 98%. Our work demonstrates the efficacy of polyamide-amine HBPs with hydrogen-bonding complementarity as adhesive binding layers, which not only toughen the interface contact but also improve efficiency, mechanical properties, and inhibit lead leakage. These findings lay the foundation for the high-efficiency, mechanically stable, and sustainable development of flexible perovskite electronics.

## Methods

### Materials

Glass/ITO ($8\,\Omega$ $sq^{-1}$) and PET/ITO ($15\,\Omega$ $sq^{-1}$) were purchased from Advanced Election Technology Co., Ltd. $SnO_2$ colloidal dispersion (15% in $H_2O$) were purchased from Alfa Aesar. MBA, ethylenediamine (EDA), butanediamine (BDA), hexanediamine (HDA) and octanediamine (ODA) were purchased from Aladdin Biochemical Technology Co., Ltd. Methylammonium chloride (MACl, 98%), formamidine iodide (FAI, 99.5%) and methylammonium bromide (MABr, 99.5%) were purchased from Great Cell Solar Corp. Lead (II) iodide ($PbI_2$, 99.99%) was purchased from TCI Shanghai Chemical Industry Materials Corp. Dimethyl formamide (DMF, 99.8%, anhydrous), dimethyl sulfoxide (DMSO, 99.9%, anhydrous), chlorobenzene(CB, 99.8%, anhydrous), 2-propanol (IPA, anhydrous, 99.5%) and acetonitrile (ACN, 99.8%, anhydrous) were purchased from Sigma-Aldrich without further purification. Methanol was purchased from Sinopharm Chemical Reagent Co., Ltd. 2,2',7,7'-tetrakis(N,N-di-p-methoxyphenylamine)-9,9-spirobifluorene (Spir-oOMeTAD, ≥99.5%), bis(trifluoromethanesulfonyl)imide (Li-TFSI, ≥99%) and 4-tert-butyl pyridine (tBP, ≥96%) were purchased from Xi'an Polymer Light Technology Corp.

### HBPs synthesis

The HBPs were synthesized by one-pot Michael addition between MBA and amine. 12.332 g of MBA was added into a round bottom flask containing mixed solvent of 60 mL of methanol and 30 mL of deionized water at 30 °C and stirred until it was dissolved totally. Then, 12.98 g of HDA was slowly dropped into the flask. The mixture was stirred at 30 °C for 24 h. After reaction, the solution was poured into a

beaker containing 1000 mL of acetone and a precipitate was produced instantaneously. The crude product was washed for three times with acetone and then dried in a vacuum oven at 30 °C for 48 h to obtain the product. The comparison samples were synthesized using the same procedure with EDA, BDA, or ODA replacing HDA.

### Device fabrication and testing

Glass/ITO and PET/ITO were cleaned in an ultrasonic bath in the sequences of detergent solution, deionized water, acetone, and isopropanol with 15 min for each step. The substrates were then treated by UV-Ozone for 20 min to improve hydrophilicity. For depositing $SnO_2$ layer, the $SnO_2$ aqueous solution (Alfa Aesar, 44592) was diluted with deionized water (1:6 volume ratio) and dispersed by ultrasonic for 30 min. The $SnO_2$ solution was spined coated on ITO at 4000 rpm for 30 s and annealed at 150 °C for 30 min. For HBP modification, HBPs solution (0.05-0.2 mg/mL in methanol) was spin-coated onto $SnO_2$ ETL at 3000 rpm for 30 s and annealed at 100 °C for 5 min in $N_2$ glovebox. $(FAPbI_3)_{1-x}(MAPbBr_3)_x$ perovskite films were prepared following the previous report. Firstly, 1.5 M $PbI_2$ dissolved in a DMF-DMSO (9:1 volume ratio) mixed solvent was spin-coated onto $SnO_2$ ETL at 1500 rpm for 30 s, followed by annealing at 70 °C for 1 min in $N_2$ glovebox. Subsequently, an FAI:MABr:MACl solution (90:9:9 mg in 1 mL IPA) was spin-coated onto the $PbI_2$ films at 2000 rpm for 30 s, followed by annealing at 150 °C for 15 min in ambient air (~40% humidity). PEAI post-treatment was applied to the perovskite films of large area and flexible devices to reduce hysteresis. PEAI solution (3 mg $mL^{-1}$ in IPA) was spin-coated on the perovskite films, followed by 10 min annealing at 100 °C. The Spiro-OMeTAD HTL was deposited on perovskite film at 3000 rpm for 30 s. The HTL solution contained 90 mg of spiro-OMeTAD doped with 28.8 µL of 4-tert-butylpyridine (tBP) and 17.5 µL of lithium bis(trifluoromethylsulphonyl)imide acetonitrile solution (LiTFSI, 520 mg/mL) in 1 mL of chlorobenzene. Finally, 100 nm Au was deposited on HTLs through thermal evaporation.

The PSC performance was tested in $N_2$ glovebox at room temperature, under AM 1.5 G, 100 mW $cm^{-2}$ illumination generated by a solar simulator (Sol3A 94063 A, Newport). The solar simulator was

calibrated using a KG-5 filtered standard solar cell (SRC-2020, NREL calibrated). The $J-V$ curves were recorded using a source meter (Keithley 2400) with a scan rate of 0.1 V s$^{-1}$ for both reverse and forward directions. The device area were defined by metal masks and set to 0.045 cm$^2$ and 1 cm$^2$ for small-size and large-size device, respectively. The incident photocurrent conversion efficiency (IPCE) spectra were measured using a quantum efficiency testing system (QE-R 3011, Enli Tech). Mott-Schottky analysis was carried out on an electrochemical workstation (Chenhua 760) with 1 kHz AC frequency and 0–1.5 V bias. Long term stability test was carried out in a N$_2$ glovebox under 100 mW cm$^{-2}$ illumination from white LED. The maximum power points of the PSCs were tracked using a solar cell stability testing system (PR-SCCS-C8Q, Puri Materials).

## Mechanical Testing
The "sandwich" double-cantilever beam (DCB) specimens were prepared with the following structure: glass/ITO/SnO$_2$/HBPs/PVK/PMMA/epoxy/ITO/glass. The dimension of the ITO-coated glass substrate used is $37.5 \times 12.5 \times 1$ mm$^3$. The deposition of SnO$_2$, HBPs, and PVK followed the aforementioned procedure. Before bonding, PMMA layer (~800 nm) was spin-coated onto the perovskite films as a barrier to the epoxy. 5-μm polytetrafluoroethylene (PTFE) film was inserted in the midplane to create a "notch" prior to epoxy application, serving as a delamination initiator. The crack initiation and propagation were monitored by high-speed photography. Subsequently, a thin layer of epoxy (~2 μm) was applied onto the PMMA layer to "glue" another cleaned ITO-coated glass substrate on top and cured overnight at room temperature in a glovebox. Excess epoxy was cleaned from the edges of the specimen before mechanical testing was conducted.

## Film characterization
The morphologies of perovskite films, fracture surfaces, and the cross-sectional of PSCs were investigated by field emission scanning electron microscopy (JSM 6700 F). The Surface morphology of the ETLs was confirmed using AFM (Bruker, Dimension Fastscan). Fourier transform infrared (FTIR) spectra were recorded in ATR mode (Thermo, Nicolet 6700). The crystal structure of perovskite was characterized using an XRD (Bruker AXS, D8 Advance). The XPS and UPS was performed using Shimadzu Kratos photoelectron spectrometers. Steady-state PL measurements were acquired using fluorescence spectrometer (FLS 980). The TRPL spectra were acquired using an Edinburgh Instruments FLS920.

## Bending durability tests
The bending durability of F-PSCs was evaluated using a mechanical tester (PR-BDM-100, Puri Materials) in constant-radius bending mode with a bending radius of 3 mm in N$_2$ glove box. The corresponding J-V curves were periodically measured at a flat state under AM 1.5 G 100 mW cm$^{-2}$ illumination.

## Lap shear tests
HBPs were placed between two identical substrates and held by clips. Lap shear testing was used to evaluate the adhesion properties, which was performed on a universal testing machine (CMT 5305). Umbonded specimens were extended at a speed of 5 mm min$^{-1}$ at 25 °C. The force was measured via a 250 N capacity load cell. The displacement was measured by a displacement sensor in the universal testing machine.

## Reporting summary
Further information on research design is available in the Nature Portfolio Reporting Summary linked to this article.

## Data availability
The source data presented in this paper is available at https://doi.org/10.6084/m9.figshare.24101076.

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

## Acknowledgements

Zhen Li acknowledges the National Key R&D Program of China (2019YFB1503201), Open project of Shannxi Laboratory of Aerospace Power (2021SXSYS-01-03), National Natural Science Foundation of China (52172238, 51902264), Natural Science Foundation of Shaanxi Province (2020JM-093), Science Technology and Innovation Commission of Shenzhen Municipality (JCYJ20190807111605472), and the Fundamental Research Funds for the Central Universities (3102019JC0005). Chao Zhang acknowledges the Fundamental Research Funds for the Central Universities (5000220118). Can Li acknowledges National Natural Science Foundation of China (52102304).

## Author contributions

Zhihao L. and Zhen L. conceived the original idea, analyzed the data, and prepared the manuscript. Zhihao L., C.J., and Z.W. carried out the fabrication, optimization, and characterization of the PSCs. J.C. carried out the mechanical tests under the supervision of J.S. and C.Z. J.X., and M.Z. carried out the finite-element simulation. C.L. assisted in the schematic drawing. All authors have given approval to the paper.

## Competing interests

The authors declare no competing interests.
