## [Peer Review File · Nature Communications]

Hyperbranched polymer functionalized flexible perovskite solar cells with mechanical robustness and reduced lead leakageREVIEWER COMMENTS

Reviewer #1 (Remarks to the Author):

The author incorporated a highly adhesive polyamide-amine-based hyperbranched polymer (HBP) into Perovskite solar cells (PSCs) and enabling a stabilized multi-layered device structure. The hydrogen bond donors and acceptors in the HBPs form a dynamic hydrogen bond network that absorbs and dissipates mechanical energy enable a robust ETL/perovskite interface, resulting in a high power conversion efficiencies of 25.05% and 23.45% for rigid and flexible PSCs, respectively. Overall, the work presents a novel dopant material and showed a competitive device performance in efficiency and mechanical stability, which was helpful to the researchers working on flexible devices. However, more details was required to support the main claims. So, I think the paper may be published if some major concerns could be addressed. And I suggest a major redivision.

Comments as below:

1. The samples were not enough to effectively evaluate the statistic results and more experiment evidence should be provided.
2. The hysteresis effect was small in small devices. However, there was still obviously hysteresis effect in large are devices and also in flexible devices, which was far legged behind that in previous reported cases and should be further optimized or make further discussion and clarification.
3. The experiment showed in Fig. 6a was not solid enough to support their claim because the cracks condition cannot be the same in different sample. Meanwhile, the ITO substrate can block water effectively and seriously influence the test results. More experiment evidence or data was required if the author still want to claim the water and/or lead leakage resistance properties.

Reviewer #2 (Remarks to the Author):

In this work, the authors employed highly branched polymers to treat SnO₂ layer to improve the adhesion between SnO₂ and perovskite layer for flexible perovskite solar cells. The work itself is interesting, it can solve this problem in some degree. However, there are still some problems unclear. Major revision are needed before final decision.

- 1.As the authors said, there are more than one intramolecular hydrogen bonding, no obvious peak shift can be seen. How the authors prove this kind of hydrogen bonding?
- 2.In the manuscript, "The polar -NH₂ end groups in the HMPs acts as an adhesive to bind with SnO₂ and perovskite layers..." However, it is needed to prove this interaction experimentally. Only schematic picture is not enough.
- 3.In Figure S6, please add the differential thermal analysis in these corresponding TG curves.
- 4.Why IR peak of HBPs at 1638 cm⁻¹ is shifted to 1645 cm⁻¹?
- 5.Although the authors provides some characterizations, however, they still told us clearly how the HBPs influenced the cell performance, or it is still unclear why the perovskite solar cells based on HBP can give the best cell performance.
- 6.As the main point of this paper is aiming at the flexible PSCs, the authors are strongly suggested to

provide enough data or characterizations of the modified SnO₂ on flexible conductive substrate in the main text, including the mechanical properties, PL, dark J-V and so on.

7. Please give the long term stability and MPP stability of flexible PSCs, at least 500 hr.

Reviewer #3 (Remarks to the Author):

This manuscript from Li et al. developed a polyamide-amine-based hyperbranched polymer (HBP) that adapts deformation and adsorbs fracture energy, increasing the toughness of the ETL/perovskite interface. The attained PSCs were at the state-of-art level with PCE of 25.05% and 23.45% for rigid and flexible PSCs. The Pb leakage reduction is also vital for practical applications, especially for wearable devices. Generally, the HBP adhesive interface layer presents an interesting approach towards robust and efficient flexible PSCs. Therefore, I would recommend this manuscript to be published in Nature communications after addressing the following comments.

1. How does the HBPs modification layer affect the conductivity of SnO₂ ETLs? It can be assumed that as an insulating polymer the HBP will increase the resistance. Does that affect the performance of PSCs.
2. How does the HBPs modification affect the contact angle of the SnO₂ ETL? And does that affect the crystallization of perovskite? Explanations and experimental results are needed.
3. The author claimed that the soft interface layer can accommodate the thermal expansion mismatch between the perovskite layer and substrate during fabrication. The authors are recommended to measure the residual stress σ_R of the interface with reference to this literature (doi: adma.202109879).
4. The HBPs adhesive interface is a typical buried interface. How does the HBPs effect the buried surfaces of the perovskite film? More experimental results should be provided.
5. Please specify and provide more details for the DCB characterization and sample preparation. For example, how is the pre-crack introduced to the specimen? How is the author certain that the samples are pulled apart from the bottom of perovskite film?
6. Although the authors demonstrate the lead sequestration ability of PSC devices with HBPs modification, The Pb²⁺ adsorption properties of the HBPs are not well characterized. They should provide some key properties for lead-absorbing, such as extraction rate and maximum absorption capacity.

Specific reply to Reviewer's comments

Title: **Enhancing mechanical stability and reducing lead leakage of flexible perovskite solar cells through a hyperbranched polymer multifunctional interface layer**

We sincerely thank all the reviewers for their time, effort and valuable comments. Point-to-point reply to the comment is listed below in blue text and corresponding revision is made in the revised manuscript and provided below in green text.

Reviewer 1:

The author incorporated a highly adhesive polyamide-amine-based hyperbranched polymer (HBP) into Perovskite solar cells (PSCs) and enabling a stabilized multi-layered device structure. The hydrogen bond donors and acceptors in the HBPs form a dynamic hydrogen bond network that absorbs and dissipates mechanical energy enable a robust ETL/perovskite interface, resulting in a high power conversion efficiencies of 25.05% and 23.45% for rigid and flexible PSCs, respectively. Overall, the work presents a novel dopant material and showed a competitive device performance in efficiency and mechanical stability, which was helpful to the researchers working on flexible devices. However, more details was required to support the main claims. So, I think the paper may be published if some major concerns could be addressed. And I suggest a major redivision.

Comments as below:

1. The samples were not enough to effectively evaluate the statistic results and more experiment evidence should be provided.

Author reply: We thank the reviewer for this valuable comment. We have added more data points to show the PCE improvement is in a reproducible way. And we have also changed the Figure 2c and Figure S16, Table S2 in the supporting information accordingly.

Figure 2c The PCE histogram of PSCs.

Figure S16. (a) V_{oc} , (b) J_{sc} , (c) FF , and (d) PCE statistics of 20 devices for the control and HBPs-modified PSCs.

Table S2. Statistical photovoltaic parameters of the devices based on SnO_2 with and without HBPs modification.

	V_{oc} (V)	J_{sc} (mA/cm ²)	FF	PCE (%)
Control	1.117±0.014	23.99±0.48	76.52±1.49	20.50±0.51
With HBPs	1.171±0.005	25.42±0.22	81.89±0.77	24.39±0.31

Revision: In the revised manuscript, we have replaced the statistics of J - V parameter and distribution of PCE as Figure S16 in the supporting information. Also, we have changed the PCE description on page 12, paragraph 2: “The average PCE of 20.50% for control devices increased to 24.39% for HBPs-modified devices”

2. The hysteresis effect was small in small devices. However, there was still obviously hysteresis effect in large area devices and also in flexible devices, which was far legged behind that in previous reported cases and should be further optimized or make further discussion and clarification.

Author reply: We thank the reviewer for this valuable comment. The origins of hysteresis are complex in PSCs. Charge accumulations at charge transport layer interface are generally considered as the main reasons for the hysteresis phenomenon. It is more difficult to attain uniform charge transport layers in large or flexible devices, because the hysteresis of the whole device is determined by the weakest part in the device. The chance of having poor interface region is higher in large-size and flexible device than in small-size rigid device. Therefore, the observed hysteresis was more obvious in the large-size and flexible devices.

Our work demonstrates that the HBPs layer serves as a passivation layer that improve charge separation at the interface and reduce hysteresis in PSC. This is evidenced by the J - V comparisons

between HBP-modified and control devices with same size or the same substrates. However, the uniformity issue still causes obvious hysteresis phenomenon in large area and flexible PSCs. Besides of the PVSK/ETL interface, PVSK/HTL interface also plays a significant role in charge separation, affecting the device performance and hysteresis. Jiang et. al. observed that PEAI treatment to the perovskite surface led to higher-efficiency cells and eliminated J-V hysteresis. The performance improvement and hysteresis elimination were related to passivation of surface defects, which reduces charge recombination and accumulation at PVSK/HTL interfaces. Unfortunately, we have not taken measurements to passivate the perovskite surface in our original submission. This could be the reason for large hysteresis in our large area and flexible devices. We have applied PEAI post-treatment to passivate the perovskite surface during revision preparation. Thanks to the surface defect passivation, we obtained large area and flexible PSCs with improved power conversion efficiency (PCE) and less hysteresis.

Revision: In the revised manuscript, we have updated the Figure 2f, Figure 5a, b, Figure S33, table S6, and the experiment details in the supporting information accordingly.

Figure 2f. J–V curve of the champion HBPs-modified PSCs with large area of 1 cm².

Figure 5. (a) J–V curves of the champion flexible devices with and without HBPs. (b) J–V curve of the champion HBPs-modified flexible PSCs with area of 1 cm².

Figure S33. V_{oc} , J_{sc} , FF, and PCE statistics of 20 flexible devices based on SnO_2 with and without HBPs modification.

Table S6. Statistical photovoltaic parameters of the flexible devices based on SnO_2 with and without HBPs modification.

ETL	V_{oc} (V)	J_{sc} (mA/cm^2)	FF (%)	PCE (%)
Control	1.094 ± 0.007	23.89 ± 0.30	76.85 ± 0.80	20.10 ± 0.34
With HBPs	1.170 ± 0.005	24.75 ± 0.60	80.50 ± 1.25	23.31 ± 0.32

In method section, the PEAI post-treatment was described:

PEAI post-treatment was applied to the perovskite films of large area and flexible devices to reduce hysteresis. PEAI solution (3 mg/mL in IPA) was spin-coated on the perovskite films, followed by 10 min annealing at 100 °C.

3. The experiment showed in Fig. 6a was not solid enough to support their claim because the cracks condition cannot be the same in different sample. Meanwhile, the ITO substrate can block water effectively and seriously influence the test results. More experiment evidence or data was required if the author still want to claim the water and/or lead leakage resistance properties.

Author reply: We thank the reviewer for this valuable comment. We agree with the reviewer that the cracks condition cannot be exactly identical in different sample. If the ITO substrate is not fully penetrated, it can also block water and influence the test results. Therefore, to avoid unintended variation in the substrate cutting, a 3×4 array of holes were intentional introduced by piercing through the flexible PSCs using a needle with diameter of 0.6 mm, as shown in Figure R1. We encapsulated the back side and edges of the flexible PSCs with 10- μm -thick PET tapes. The PSCs

with through holes on the front side were then immersed in 200 mL of deionized water to simulate lead leakage under immersion conditions. As depicted in Figure R2, yellow color appeared in the control device surrounding the needle-pierced holes and expanded in circular shapes within a few minutes. The yellow region expanding was slower in the PET/ITO region than in the PET-only region, indicating that ITO itself prevents water lateral penetration at a certain extent. After 60 minutes, the yellow region expanded to most of the device area. In contrast, the device modified with HBPs only exhibited yellowing around the pierced hole area, and the yellow region did not spread outward. The lead concentration in water was determined using inductively coupled plasma mass spectrometry (ICP-MS). Within 30 minutes, the Pb concentration in the polluted water with the control device increased significantly due to lateral water penetration and the rapid dissolution of lead compounds. Remarkably, the device with HBPs modification retained a lead sequestration efficiency of 98% compared to the control device, which is similar to the results in our cutting experiment.

Figure R1. The schematics of damaged flexible PSCs piercing with a needle to form an array for lead leakage test. (b) Pb concentration in the contaminated water measured by ICP-MS.

Subsequently, we have also investigated the intrinsic lead-absorption properties of the HBPs. Firstly, 2 mL Pb^{2+} solution (10 ppm) was added to the bottles with certain amount of HBPs or polyamide (PA) absorbents. The bottles were then stirred for 30 min to reach equilibrium and the solution was collected for ICP-MS analysis to measure the remaining Pb^{2+} concentration. As shown in Figure S45a, the extraction efficiency increased with increased amounts of HBPs or PA. When the weight of absorbent reaches 6 mg, the extraction amount already reaches 97.5% for HBPs higher than 72.2% for PA.

The lead-adsorbing ability of PA and HBPs film was also quantified by the absorption kinetics measurements. For kinetics measurements, 1.5×1.5 cm² PA/Glass and HBPs/Glass samples (net weight of PA and HBPs film was all fixed to 6 mg) were placed facing up into the PbI_2 aqueous solution (10 ppm, 50.0 mL) with agitational stirring. The solution was collected from the bottle with

different time intervals from 0 to 360 min and then quantitative analyzed by ICP-MS to determine the lead concentration. The amount of Pb adsorbed per unit mass (q_t) of film at the time (t) was determined by the expression $q_t = \frac{V(C_0 - C_t)}{m}$, where C_0 is the initial concentration, C_t is the concentration at time (t), V is the volume of the solution, and m is the mass of the absorbent. The adsorption kinetics were analyzed using a pseudo-second-order model: $q_t = \frac{tK_2q_e^2}{1+tK_2q_e}$.

As shown in Figure S45b, HBPs film adsorbs most of the Pb^{2+} (90%) within 30 min at room temperature, but PA film only adsorbs 55% of the Pb^{2+} at the same time duration. The adsorption kinetics of Pb^{2+} were fitted by a pseudo-second-order model (Figure S45c) based on the data from Figure S45b. The sorption rate constant (K_2) of HBPs is $8.38 \text{ mg min}^{-1} \text{ g}^{-1}$, which is higher than $4.75 \text{ mg min}^{-1} \text{ g}^{-1}$ of PA, and the adsorption capacity at equilibrium (q_e) of HBPs toward Pb^{2+} is 78.49 mg g^{-1} , higher than 53.53 mg g^{-1} of PA. The higher adsorption capacity is due to the internal cavity of the HBPs and high density functional groups. The faster adsorption rate and higher adsorption capacity suggest that the HBPs film can sequence the leaked Pb^{2+} from damaged PSCs more effectively.

Figure S45. Pb^{2+} absorption properties of different absorbents. (a) Influence of absorbent weight toward Pb^{2+} extraction efficiency. (b) Pb^{2+} sorption kinetics and the corresponding adsorbed Pb^{2+} amount of PA and HBPs films. (c) The kinetics fitting curves of PA and HBPs films from a pseudo-second-order mode.

Revision: In the revised manuscript, we have changed the Figure 6a, Figure 6b and Figure S46 in the supporting information accordingly. Meanwhile, we have added the Pb extraction behavior of PA and HBPs films as Figure S45 in the supporting information. Also, have added a sentence on page 25, paragraph 2: “We investigated the Pb^{2+} absorption properties of HBPs. As shown in Figure S45, the HBPs film showed a saturated adsorption capacity of 78.49 mg g^{-1} , higher than 53.53 mg g^{-1} of linear polymer PA. The adsorption rate of HBPs film was $8.38 \text{ mg min}^{-1} \text{ g}^{-1}$ also higher than $4.75 \text{ mg min}^{-1} \text{ g}^{-1}$ of PA. The better Pb^{2+} absorption properties can be attributed to the abundant internal cavity of HBPs.”

Reviewer #2 (Remarks to the Author):

In this work, the authors employed highly branched polymers to treat SnO_2 layer to improve the adhesion between SnO_2 and perovskite layer for flexible perovskite solar cells. The work itself is interesting, it can solve this problem in some degree. However, there are still some problems unclear. Major revision are needed before final decision.

Author reply: We thank the reviewer for appreciating the significance of our study and providing valuable comments. We have addressed his/her comments and believe that the revision has led to an improved manuscript.

1. As the authors said, there are more than one intramolecular hydrogen bonding, no obvious peak shift can be seen. How the authors prove this kind of hydrogen bonding?

Author reply: To characterize multiple hydrogen-bonding interactions of HBPs, temperature-dependent FTIR measurement was performed. Obviously, there are three peaks centered at 1636, 1532 and 3273 cm^{-1} at room temperature, belonging to the hydrogen-bonded amide I band, -NH- bending and -NH₂ bending, respectively, as shown in Figure S8. Moreover, these peaks are very broad, indicating the coexistence of multiple hydrogen bonds and free amide, -NH- and -NH₂ groups in the HBPs molecules. As the temperature increases from 25 to 175 °C, the intensity around 1636, 1532 and 3273 cm^{-1} decreases, while the intensity around 1644 cm^{-1} (free amide groups), 1522 cm^{-1} (free -NH- groups) and 3284 cm^{-1} (free -NH₂ groups) increases remarkably. The results show that as temperature increases, the associated hydrogen bonds gradually become dissociated. Such shifting is observed at temperature as low as 50 °C, indicating that part of the groups already forms dynamic hydrogen bonds at room temperature, which is consistent with our DSC results (Figure S5).

Revision: In the revised manuscript, we have added the temperature-dependent FTIR spectra HBPs as Figure S8 in the supporting information. Also, we discussed the FTIR spectra in the manuscript (page 9, paragraph 1) and it is also given below:

The dynamic hydrogen bond is also demonstrated by FTIR band shift of the amide, -NH- and -NH₂ groups in associated state to free state at raising temperature from room temperature to 175 °C (Figure S8).

Figure S8. Changes of hydrogen bonds in HBPs at raising temperature. temperature-dependent FTIR spectra of HBPs upon heating from 25 to 175 °C, spectra at 25, 50, 75, 100, 125, 150 and 175 °C are presented with color from blue to red.

2. In the manuscript, "The polar -NH₂ end groups in the HMPs acts as an adhesive to bind with SnO₂ and perovskite layers..." However, it is needed to prove this interaction experimentally. Only schematic picture is not enough.

Author reply: We thank the reviewer for this valuable comment. Adhesives with high-density amino groups are reported to improve adhesive strength, which is highly desirable for bonding metal, wood, and engineering plastics (Materials Science & Engineering C, 2010, 111, 110796; Progress

in Polymer Science 2021, 116, 101388). Therefore, we assumed the abundant polar terminal amino groups in the hyperbranched polyamide-amine can form high-density multiple hydrogen bonds with the surface, providing a great adhesive strength. We have taken X-ray photoelectron spectroscopy (XPS) and Fourier-transform infrared (FTIR) spectroscopy characterizations to gain insight into the chemical interactions between HBPs and the adjacent surface. Firstly, with HBPs modification, the binding energies of the Sn 3d peaks (Figure 1e) of SnO₂ were upshifted. The FTIR peak of –NH₂ (Figure S12d) at 3305 cm⁻¹ was shifted to 3299 cm⁻¹. Both of the shifts indicate a strong chemical interaction between HBPs and SnO₂. The improved adhesion strength can be assigned to hydrogen bonding between the polar –NH₂ groups and the SnO₂ surface (Adv. Mater. 2022, 34, 2106118). Secondly, the chemical interaction between HBPs and perovskite was also characterized. The Lewis acid-base coordination between amino group and PbI₂ was widely reported in the modification of perovskite (Sci. Adv. 2019; 5: eaav8925, Adv. Funct. Mater. 2022, 32, 2204725). We observed the Pb 4f peaks was shifted to higher binding energy (Figure S13a). In addition, the FTIR spectrum (Figure 1f) revealed that the stretching vibration band of C=N and -NH in perovskite formed hydrogen bonds with the –NH– and –NH₂ groups of HBPs, resulting in a slight shift of about 15 cm⁻¹ toward lower wavenumber. Therefore, it can be inferred that HBPs can form chemical interactions with both SnO₂ and perovskite layers through hydrogen bonds and coordination bonds, thus acting as a bridge between ETL and perovskite and strengthen the interface. We have also provided the lap-shear tests (Figure S29), nano scratch tests (Figure 4a) and double cantilever beam (DCB) (Figure 4c) measurements to quantify the adhesion strength between the SnO₂ and the perovskite layers in the manuscript to show the strong interface bonding induced by HBPs.

Revision: In the revised manuscript, we have added some reference to analysis functional properties of polyamide adhesives and added a sentence on page 9, paragraph 1 to emphasize the adhesive mechanism with amino group.

“The polar -NH₂ end groups in the HBPs can form high-density multiple hydrogen bonds with SnO₂ and perovskite layers, which are beneficial to provide greater adhesive strength.”

3. In Figure S6, please add the differential thermal analysis in these corresponding TG curves.

Author reply/revision: As suggested, we have added differential thermal analysis in the corresponding TG curves in the revised supporting information.

Figure S6. Thermogravimetric analysis (TGA) curves of HBPs with (a) EDA, (b) BDA, (c) HDA and (d) ODA

4. Why IR peak of HBPs at 1638 cm^{-1} is shifted to 1645 cm^{-1} ?

Author reply: As we mentioned in the reply to comment 1, the $-\text{NH}-$, $-\text{NH}_2$ and $\text{C}=\text{O}$ groups can form strong hydrogen bonds between HBPs and SnO_2 , therefore the $\text{C}=\text{O}$ peak of HBPs at 1638 cm^{-1} shifted to 1645 cm^{-1} .

5. Although the authors provides some characterizations, however, they still told us clearly how the HBPs influenced the cell performance, or it is still unclear why the perovskite solar cells based on HBP can give the best cell performance.

Author reply: We thank the reviewer for this valuable comment. It has been reported that there are plenty of oxygen vacancies and unsaturated Sn dangling bonds in the SnO_2 film, which strongly affect the performance of PSCs. When HBPs is introduced on the surface of the SnO_2 film, the $-\text{C}=\text{O}$, $-\text{NH}-$, and $-\text{NH}_2$ groups of HBPs can passivate surface defects, which is verified by our FTIR and XPS results (Figure S11, 12). We have carried out conductivity measurement to the SnO_2 layer with interdigital electrodes (Figure R2). The conductivity is normalized to the electrode geometry and calculated according to the Equation:

$$\sigma = \frac{I}{V} \frac{d}{(2n-1)lh}$$

Where I is the measured current, V is the applied voltage, d (0.2 mm) is the spacing between adjacent electrodes, n (6) is the number of finger pairs, l (5 mm) is the length of the overlap area of the fingers, and h (30 nm) is the thickness of the SnO_2 ETL film. Since the HBP layer is very thin in our device, the interface layer does not show negative affect to the ETL conductivity. The conductivity is even slightly improved after HBP modification, probably due to the n-doping effect of the $-\text{NH}_2$ groups in HBPs.

Moreover, the inserted insulating layer can serve as a tunneling layer at interface, which have been shown to suppress carrier back flow and passivate the interface to improve PSC efficiency. The tunneling layer can selectively conduct one type of charges while block the other, which spatially separates photogenerated electrons and holes (Adv. Mater. 2016, 28, 6734–6739; Adv. Funct. Mater. 2019, 29, 1905336). The HBPs is inserted between the perovskite and the SnO₂ electron layers, and the energy diagram is shown in Figure R3. The thin insulating layer allows the transport of photogenerated electrons from perovskite to SnO₂ layer by tunneling, and also blocks the back flow of electrons into the perovskite layer. Moreover, the perovskite film with a HBPs layer shows much lower interface defect density than the control film (Figure 3d). The oxygen vacancies in the SnO₂ film can be also reduced when HBPs is introduced on the surface of the SnO₂ film. Thus, interface recombination is suppressed by HBPs. Overall, the HBPs layer can facilitate carrier separation and suppress interface recombination at the perovskite and ETL interface, improving PCE of the PSCs. We have conducted transient photocurrent (TPC) decay and transient photovoltage (TPV) decay measurements to investigate the charge transfer and charge extraction efficiency between the perovskite films and the SnO₂ layers. As shown in Figure. R4, the time constant of photocurrent decay for HBP modified PSCs is 0.60 μ s, shorter than 0.77 μ s for control device, indicating faster charge transport. Meanwhile the TPV lifetime for devices with HBPs modification is about 2.18 ms, substantially longer than 1.05 ms for the control devices. The longer charge recombination lifetime from TPV decay can be linked to slower interfacial charge recombination at the HBPs modified interface.

Secondly, the influence of HBPs modification to the PSCs is also investigated. The PVSK/SnO₂ interface is a typical buried interface as commented by our reviewer #3. Therefore, we conducted more characterizations to the buried PVSK/SnO₂ interface by peeling off the perovskite layer to expose the interface layer using a reported sample preparing method (doi.org/10.1038/s41566-023-01247-4). As shown in Figure S26, the buried interface of the HBPs-modified film has a higher surface potential than the control film, implying that carriers can transport through the buried interface to the ETL more efficiently. Furthermore, conductive atomic force microscopy also directly demonstrated that the buried interface of the HBPs-modified film has drastically elevated conductivity.

Lastly, we have further investigated residual stress within the perovskite/SnO₂ film stacks without and with the HBPs by grazing incident X-ray diffraction (GIXRD) with $d_{(hkl)}\text{-sin}^2(\Psi)$ method (Figure S24). The HBPs modification layer could reduce \sim 56% residual tensile strain with respect to the control sample, which improves efficiency and stability in corresponding PSCs.

Figure R2. Conductivity measurement of SnO₂ and HBP-modified-SnO₂ films.

Figure R3. Schematic illustration for carrier transportation processes related to the HBP buffer layer passivating interface recombination: (a) the device without HBP buffer layer and (b) the device with HBP buffer layer.

Figure R4. Normalized TPC (a) and TPV (b) for PSCs without and with HBP modification.

Figure S26. (a) Schematic of process to expose the buried interface and (b) corresponding sample photos. (c) Conductive atomic force microscopy images of the buried interface in control and target devices. (d) Surface potential distribution of the buried interface in control and target device

measured by Kelvin probe force microscopy. (e) Schematic diagram of control and target buried interface.

Figure S24. Residual stress of perovskite films. GIXRD patterns at different Ψ angles (from 0° to 55°) for perovskite films formed on SnO_2 ETLs (a) control and (b) with HBPs modification, respectively. (c) Lattice spacing $d_{(012)}$ versus $\sin^2(\Psi)$ plots for perovskite films formed on SnO_2 ETLs without and with HBPs, respectively. (d) Residual stress σ_R comparison between control and HBPs-modified perovskite films.

Revision: In the revised manuscript, we have added the GIXRD spectra of control and HBPs-modified perovskites films as Figure S24 in the supporting information. Also, we discussed the GIXRD in the manuscript (page 14, paragraph 2) and it is also given below:

Furthermore, the residual tensile strain of perovskite films was measured by grazing incident X-ray diffraction (GIXRD). The residual tensile strain of HBPs-modified was 28.30 MPa, constituting ~56% strain reduction compared to 64.24 MPa of the control sample (Figure S24).

In addition, we also discussed the optoelectronic properties of buried interfaces in the manuscript (page 15, paragraph 2) and it is also given below:

We further investigate the electrical properties of the buried interfaces (Figure S26). Higher surface potential and elevated conductivity of the buried interface rigorously shows that the HBPs layer can improve the carrier extraction at the ETL/perovskite interface.

6. As the main point of this paper is aiming at the flexible PSCs, the authors are strongly suggested to provide enough data or characterizations of the modified SnO_2 on flexible conductive substrate in the main text, including the mechanical properties, PL, dark J-V and so on.

Author reply: We thank the reviewer for this valuable comment. We have provided the XRD patterns of perovskite films deposited on flexible substrates based on SnO_2 and HBPs-modified SnO_2 before and after bending to gain insights into the effects of the HBPs modification on structural

defects by Williamson–Hall (WH) analysis (Figure S34, S35). And we have also conducted in-situ steady-state photoluminescence (PL) measurements on the perovskite films with bending test (Figure 5d, e) and tensile test (Figure S36, 37) to further elucidate the effects of HBPs on the material composition stability due to mechanical stress. Furthermore, based on the measured mechanical properties, we performed finite element simulation to illustrate HBPs modification can improve the mechanical durability of the device (Figure 5f). In addition, we also evaluated the mechanical stability of F-PSCs at various curvature radii (Figure 5g) and carried out fatigue tests with a curvature radius of 3 mm and 10000 bending cycles (Figure 5h). As suggested, we would like to take this opportunity to add more data of the flexible PSCs in the supporting information. Firstly, we have investigated the SnO₂ surface morphologies on flexible PET substrate (Figure R5). The surface roughness (RMS) of SnO₂ ETLs was reduced from 1.12 nm (control) to 1.06 nm (HBPs-modified), which is consistent with the results on rigid glass substrates. Meanwhile, we also conducted dark *J-V* measurement of flexible PSCs with and without HBPs and found the HBPs-modified device exhibited lower leakage current, indicating it enhances the hole-blocking ability and suppresses charge recombination at the ETL/perovskite interface (Figure R6). In addition, we have conducted the SCLC measurements of control and HBPs-modified perovskites films after bending test, and found the trap density in control sample (42%) had a huge increase compared to the HBPs-modified sample (11%) (Figure S38).

Revision: In the revised manuscript, we have added the SCLC measurements of control and HBPs-modified perovskites films after bending test as Figure S38 in the supporting information. Also, we discussed the SCLC measurements in the manuscript (page 22) and it is also given below:

Meanwhile, the trap density in control sample (42%) had a huge increase compared to the HBPs-modified sample (11%) (Figure S38).

Figure R5. AFM images of (a) SnO₂ and (b) HBPs-modified SnO₂ films on the ITO/PET flexible substrates.

Figure R6. Dark J - V curves of flexible PSCs with and without HBPs modification.

Figure S38. SCLC measurements of perovskite films deposited on (a) SnO_2 and (b) HBPs-modified SnO_2 before and after bending with the electron-only structures. (c) Comparison of trap density of control and HBPs-modified perovskite films before and after bending.

7. Please give the long term stability and MPP stability of flexible PSCs, at least 500 hr.

Author reply: We thank the reviewer for this valuable comment. We have provided the long-term environmental stability in ambient air with 30-40% relative humidity (Figure S42) and long-term light stability under continuous 1-sun illumination in N_2 environment of the unencapsulated flexible PSCs in the original manuscript (Figure S43). We have also measured the MPP stability of the flexible PSCs as suggested.

Revision: In the revised manuscript, we have added the MPP stability of flexible PSCs as Figure S44 in the supporting information. Also, we discussed the MPP stability in the manuscript (page 25) and it is also given below:

For the long-term operational stability, the efficiency of the control PSC decays to ~30% of the initial value, while the HBPs-modified PSC maintains 91% of its initial efficiency after continuous operation for 500 h (Figure S44).

Figure S44. The unencapsulated F-PSCs with and without HBPs modification under MPP tracking and one-sun illumination in N_2 at 30°C

Reviewer #3 (Remarks to the Author):

This manuscript from Li et al. developed a polyamide-amine-based hyperbranched polymer (HBP)

that adapts deformation and adsorbs fracture energy, increasing the toughness of the ETL/perovskite interface. The attained PSCs were at the state-of-art level with PCE of 25.05% and 23.45% for rigid and flexible PSCs. The Pb leakage reduction is also vital for practical applications, especially for wearable devices. Generally, the HBP adhesive interface layer presents an interesting approach towards robust and efficient flexible PSCs. Therefore, I would recommend this manuscript to be published in Nature communications after addressing the following comments.

Author reply: We thank the reviewer for appreciating the significance of our study and providing valuable comments. We have addressed his/her comments and believe that the revision has led to an improved manuscript.

1. How does the HBPs modification layer affect the conductivity of SnO₂ ETLs? It can be assumed that as an insulating polymer the HBP will increase the resistance. Does that affect the performance of PSCs.

Author reply: We thank the reviewer for this valuable comment. we have carried out conductivity measurement to the SnO₂ layer with interdigital electrodes (Figure R2). The conductivity is normalized to the electrode geometry and calculated according to the Equation:

$$\sigma = \frac{I}{V} \frac{d}{(2n - 1)lh}$$

Where I is the measured current, V is the applied voltage, d (0.2 mm) is the spacing between adjacent electrodes, n (6) is the number of finger pairs, l (5 mm) is the length of the overlap area of the fingers, and h (30 nm) is the thickness of the SnO₂ ETL film. Since the HBP layer is very thin in our device, the interface layer does not show negative affect to the ETL conductivity. The conductivity is even improved after HBP modification slightly, probably due to the n-doping effect of the -NH₂ groups in HBPs.

The inserted insulating layer can serve as a tunneling layer at interface, which have been shown to suppress carrier back flow and passivate the interface to improve PSC efficiency. The tunneling layer can selectively conduct one type of charges while block the other, which spatially separates photogenerated electrons and holes (Adv. Mater. 2016, 28, 6734–6739; Adv. Funct. Mater. 2019, 29, 1905336). The HBPs is inserted between the perovskite and the SnO₂ electron layers, and the energy diagram is shown in Figure R3. The thin insulating layer allows the transport of photogenerated electrons from perovskite to SnO₂ layer by tunneling, and also blocks the back flow of electrons into the perovskite layer. Moreover, the perovskite film with a HBPs layer shows much lower interface defect density than the control film (Figure 3d). The oxygen vacancies in the SnO₂ film can be also reduced when HBPs is introduced on the surface of the SnO₂ film. Thus interface recombination is suppressed by HBPs. Overall, the HBP layer can facilitate carrier separation and suppress interface recombination at the perovskite and ETL interface, improving PCE of the PSCs. Not surprisingly, the performance of PSCs will certainly decrease with with a thick insulating polymer. Therefore, we have optimized the concentration of HBPs to achieve the best PCEs (Figure S14).

Figure R2. Conductivity measurement of SnO₂ and HBP-modified-SnO₂ films.

Figure R3. Schematic illustration for carrier transportation processes related to the HBP buffer layer passivating interface recombination: (a) the device without HBP buffer layer and (b) the device with HBP buffer layer.

2. How does the HBP modification affect the contact angle of the SnO₂ ETL? And does that affect the crystallization of perovskite? Explanations and experimental results are needed.

Author reply: We thank the reviewer for this valuable comment. Usually, the surface energy of the substrate affects the morphology and crystallinity of the resulting perovskite films. Therefore, we conducted contact angle measurement. From figures R7, we can see that the contact angle of HBP modified SnO₂ films decreases from 71.1° to 41.2°, which helps to increase the nucleation density, promoting nucleation and growth process of perovskite films, and ultimately lead to the formation of larger grain size perovskite films.

It is also assumed that the chemical affinity of Pb²⁺ toward amide and amino groups improve the crystallinity of the film. This chemical interaction is shown by the FTIR spectra of the HBP and HBP-PbI₂ mixture. As shown in Figure R8, the characteristic peak of amide group in FTIR spectra shift from 1534 and 1637 cm⁻¹ for HBP to 1522 and 1626 cm⁻¹ for HBP-PbI₂.

Figure R7. Contact angle measurements of perovskite precursors on (a) ITO/SnO₂ and (b) ITO/HBPs-modified SnO₂.

Figure R8. FTIR spectra of HBPs and HBPs-PbI₂.

3. The author claimed that the soft interface layer can accommodate the thermal expansion mismatch between the perovskite layer and substrate during fabrication. The authors are recommended to measure the residual stress σ_R of the interface with reference to this literature (doi: adma.202109879).

Author reply: We thank the reviewer for this valuable comment. we have measured the residual stress at the perovskite/SnO₂ interface by grazing incident X-ray diffraction (GIXRD) with $d_{(hkl)} \cdot \sin^2(\Psi)$ method.

As illustrated in Figure S24a, by varying Ψ angles from 0° to 55°, the peak for (012) plane of perovskite without HBPs additive shifts gradually to smaller 2θ . The corresponding lattice spacing $d_{(012)}$ increased monotonically (Figure S24c), thus indicating increasing tensile stress within the perovskite film from the free surface to the ETL. Meanwhile, HBPs-treated perovskite film counterpart exhibits negligible shift of lattice spacing regardless of the X-ray incident angle Ψ (Figure S24b), suggesting that the lattice spacing $d_{(012)}$ is almost unchanged at different depths of the perovskite film. The constant lattice spacing indicates that the thermal induced residual stress is released with the soft HBP interface layer. The residual tensile stress of the control and HBPs-modified perovskite films are calculated to be 64.24 MPa and 28.30 MPa, respectively, constituting $\approx 80\%$ reduction of the residual tensile stress at the interface (Figure S24d). The reduced stress is beneficial to the efficiency and stability of resulting F-PSCs.

This interface stress release is due to the low glass transition temperature of HBPs (around 40 °C), derived from the differential scanning calorimetry (DSC) measurement (Figure S5). As the annealing temperature of the perovskite is 150 °C, much higher than the glass transition temperature of the HBPs, the HBPs layer remains in a soft state when the perovskite film is annealing. The soft interface layer prevents the direct contact with the rigid ETL layer while the perovskite crystal grows. Therefore, the perovskite with a smaller interface stress can be obtained when HBPs is introduced onto the surface of the SnO₂, improving the performance and stability of PSCs during working condition (Figure S42, 43 and Figure S44).

Revision: In the revised manuscript, we have added the GIXRD of control and HBPs-modified perovskites films as Figure S24 at in the supporting information. Also, we discussed the GIXRD in

the manuscript (page 14) and it is also given below:

Furthermore, the residual tensile strain of perovskite films was measured by grazing incident X-ray diffraction (GIXRD). The residual tensile strain of HBPs-modified was 28.30 MPa, constituting ~56% strain reduction compared to 64.24 MPa of the control sample (Figure S24).

Figure S24. Residual stress of perovskite films. GIXRD patterns at different Ψ angles (from 0° to 55°) for perovskite films formed on SnO₂ ETLs (a) control and (b) with HBPs modification, respectively. (c) Lattice spacing $d_{(012)}$ versus $\text{sin}^2(\Psi)$ plots for perovskite films formed on SnO₂ ETLs without and with HBPs, respectively. (d) Residual stress σ_R comparison between control and HBPs-modified perovskite films.

4. The HBPs adhesive interface is a typical buried interface. How does the HBPs effect the buried surfaces of the perovskite film? More experimental results should be provided.

Author reply: We thank the reviewer for this valuable comment. As we mentioned in the reply to comment 2, HBPs modification reduce the surface energy and contact angle of the SnO₂ surface, promoting the nucleation and growth of perovskite grains, and ultimately lead to the formation of larger grain in the final perovskite films. The PVSK/SnO₂ interface is a typical buried interface. Therefore, we conducted more characterizations to the buried PVSK/SnO₂ interface by peeling off the perovskite layer to expose the interface layer using a reported sample method (doi.org/10.1038/s41566-023-01247-4). As shown in Figure S26, the buried interface of the HBPs-modified film has a higher surface potential than the control film, implying that carriers can transport through the buried interface to the ETL more efficiently. Furthermore, conductive atomic force microscopy also directly and visually demonstrated that the buried interface of the HBPs-modified film has drastically elevated conductivity (Figure S26).

Revision: In the revised manuscript, we have added the Kelvin probe force microscopy (KPFM) and conductive atomic force microscopy (c-AFM) of control and HBPs-modified perovskites films

as Figure S26 in the supporting information. Also, we discussed the KPFM and c-AFM spectra in the manuscript (Page 15) and it is also given below:

We further investigate the electrical properties of the buried interfaces (Figure S26). Higher surface potential and elevated conductivity of the buried interface rigorously shows that the HBPs layer can improve the carrier extraction at the ETL/perovskite interface.

Figure S26. (a) Schematic of process to expose the buried interface and (b) corresponding sample photos. (c) Conductive atomic force microscopy images of the buried interface in control and target devices. (d) Surface potential distribution of the buried interface in control and target device measured by Kelvin probe force microscopy. (e) Schematic diagram of control and target buried interface.

5. Please specify and provide more details for the DCB characterization and sample preparation. For example, how is the pre-crack introduced to the specimen? How is the author certain that the samples are pulled apart from the bottom of perovskite film?

Author reply: We thank the reviewer for this valuable comment. The double cantilever beam (DCB) test was conducted according to the protocol for measuring mode I interlaminar fracture toughness G_{IC} for unidirectionally reinforced materials. As shown in Figure R9, the sample consists of a rectangular, uniform thickness sample containing a non-adhesive insert on the midplane that serves as a delamination initiator. To create a “notch”, 5-mm non-adhesive PTFE film was insert on the midplane that serves as a delamination initiator prior to epoxy application. And the crack initiation was monitored by high-speed photography, which can guarantee interlaminar fracture using DCB specimens at intermediate displacement.

The photograph of perovskite film pulled apart from bottom substrate was shown in Figure S30c. Due to the strong adhesive of HBPs, the perovskite film was pulled apart from the HBPs-modified SnO_2 substrate, leaving obvious residual perovskite grains on the bare SnO_2 film. In addition, SEM images of the delaminated surface of SnO_2 ETL and perovskite after DCB tests are shown in Figure 4f and 4g, respectively. Some small perovskite grains were observed on the HBPs-modified SnO_2

surface, while the perovskite surface with HBPs modification exhibited some holes left by the perovskite grains that attached to the SnO₂ surface, suggesting the samples are pulled apart from the bottom of perovskite film.

Figure R9. Photograph of pre-crack introduced to the double cantilever beam (DCB) sample

Revision: In the revised manuscript, we specify the details of the introduction of pre-crack into the specimen in the mechanical testing section of the method section.

5- μm polytetrafluoroethylene (PTFE) film was inserted in the midplane to create a “notch” prior to epoxy application, serving as a delamination initiator. The crack initiation and propagation were monitored by high-speed photography.

6. Although the authors demonstrate the lead sequestration ability of PSC devices with HBPs modification, The Pb²⁺ adsorption properties of the HBPs are not well characterized. They should provide some key properties for lead-absorbing, such as extraction rate and maximum absorption capacity.

Author reply: We thank the reviewer for this valuable comment. We have also investigated the intrinsic lead-absorption properties of the HBPs. Firstly, 2 mL Pb²⁺ solution (10 ppm) was added to the bottles with certain amount of HBPs or polyamide (PA) absorbents. The bottles were then stirred for 30 min to reach equilibrium and the solution was collected for ICP-MS analysis to measure the remaining Pb²⁺ concentration. As shown in Figure S45a, the extraction efficiency increased with increased amounts of HBPs or PA. When the weight of absorbent reaches 6 mg, the extraction amount already reaches 97.5% for HBPs higher than 72.2% for PA.

The lead-adsorbing ability of PA and HBPs film was also quantified by the absorption kinetics measurements. For kinetics measurements, 1.5 \times 1.5 cm² PA/Glass and HBPs/Glass samples (net weight of PA and HBPs film was all fixed to 6 mg) were placed facing up into the PbI₂ aqueous solution (10 ppm, 50.0 mL) with agitational stirring. The solution was collected from the bottle with different time intervals from 0 to 360 min and then quantitative analyzed by ICP-MS to determine the lead concentration. The amount of Pb adsorbed per unit mass (q_t) of film at the time (t) was determined by the expression $q_t = \frac{V(C_0 - C_t)}{m}$, where C_0 is the initial concentration, C_t is the concentration at time (t), V is the volume of the solution, and m is the mass of the absorbent. The

adsorption kinetics were analyzed using a pseudo-second-order model: $q_t = \frac{tK_2q_e^2}{1+tK_2q_e}$.

As shown in Figure S45b, HBPs film adsorbs most of the Pb²⁺ (90%) within 30 min at room

temperature, but PA film only adsorbs 55% of the Pb^{2+} at the same time duration. The adsorption kinetics of Pb^{2+} were fitted by a pseudo-second-order model (Figure S45c) based on the data from Figure S45b. The sorption rate constant (K_2) of HBPs is $8.38 \text{ mg min}^{-1} \text{ g}^{-1}$, which is higher than $4.75 \text{ mg min}^{-1} \text{ g}^{-1}$ of PA, and the adsorption capacity at equilibrium (q_e) of HBPs toward Pb^{2+} is 78.49 mg g^{-1} , higher than 53.53 mg g^{-1} of PA. The higher adsorption capacity is due to the internal cavity of the HBPs and high density functional groups. The faster adsorption rate and higher adsorption capacity suggest that the HBPs film can sequence the leaked Pb^{2+} from damaged PSCs more effectively.

Figure S45. Pb^{2+} absorption properties of different absorbents. (a) Influence of absorbent weight toward Pb^{2+} extraction efficiency. (b) Pb^{2+} sorption kinetics and the corresponding absorbed Pb^{2+} amount of PA and HBPs films. (c) The kinetics fitting curves of PA and HBPs films from a pseudo-second-order mode.

Revision: In the revised manuscript, we have added the Pb extraction behavior of PA and HBPs films as Figure S45 in the supporting information. Also, have added a sentence on page 25, paragraph 2: “We investigated the Pb^{2+} absorption properties of HBPs. As shown in Figure S45, the HBPs film showed a saturated adsorption capacity of 78.49 mg g^{-1} , higher than 53.53 mg g^{-1} of linear polymer PA. The adsorption rate of HBPs film was $8.38 \text{ mg min}^{-1} \text{ g}^{-1}$ also higher than $4.75 \text{ mg min}^{-1} \text{ g}^{-1}$ of PA. The better Pb^{2+} absorption properties can be attributed to the abundant internal cavity of HBPs.”

REVIEWERS' COMMENTS

Reviewer #1 (Remarks to the Author):

All my comments have been well addressed and I recommend to accept the work.

Reviewer #2 (Remarks to the Author):

The authors have already revised the manuscript based on the Reviewer's comments. It can be acceptable in Nat. Commun. without further revision.

Reviewer #3 (Remarks to the Author):

All questions or concerns have been well addressed, enabling the manuscript reaching the level of NC. I'd like to recommend it for publication on NC as its presents form.